# CD26-negative and CD26-positive tissue-resident fibroblasts contribute to functionally distinct CAF subpopulations in breast cancer

Julia M. Houthuijzen [1] ✉, Roebi de Bruijn[1,2], Eline van der Burg[1], Anne Paulien Drenth[1], Ellen Wientjens[1], Tamara Filipovic[1], Esme Bullock[3], Chiara S. Brambillasca[1], Emilia M. Pulver [1], Marja Nieuwland[4], Iris de Rink[4], Frank van Diepen[5], Sjoerd Klarenbeek [6], Ron Kerkhoven[4], Valerie G. Brunton [3], Colinda L.G.J. Scheele [7,8], Mirjam C. Boelens[1] & Jos Jonkers [1] ✉

Cancer-associated fibroblasts (CAFs) are abundantly present in the micro-environment of virtually all tumors and strongly impact tumor progression. Despite increasing insight into their function and heterogeneity, little is known regarding the origin of CAFs. Understanding the origin of CAF heterogeneity is needed to develop successful CAF-based targeted therapies. Through various transplantation studies in mice, we show that CAFs in both invasive lobular breast cancer and triple-negative breast cancer originate from mammary tissue-resident normal fibroblasts (NFs). Single-cell transcriptomics, in vivo and in vitro studies reveal the transition of CD26+ and CD26- NF populations into inflammatory CAFs (iCAFs) and myofibroblastic CAFs (myCAFs), respectively. Functional co-culture experiments show that CD26+ NFs transition into pro-tumorigenic iCAFs which recruit myeloid cells in a CXCL12-dependent manner and enhance tumor cell invasion via matrix-metalloproteinase (MMP) activity. Together, our data suggest that CD26+ and CD26- NFs transform into distinct CAF subpopulations in mouse models of breast cancer.

Tumorigenesis is not only governed by genetically altered cancer cells per se, but also by non-malignant host cells in the tumor microenvironment (TME), which strongly impact tumor progression, metastasis, and therapy response (reviewed in ref. [1]). One of the most dominant cell types within the TME are the cancer-associated fibroblasts (CAFs). Decades of research has shown that CAFs can affect all hallmarks of cancer (reviewed in ref. [2]). However, to consider CAFs solely as tumor-promoting within the TME is too simplistic. Previous studies have also identified CAFs with tumor-restraining properties[3,4]. Multiple studies have uncovered CAFs as a heterogeneous cell population, an observation that has been further highlighted by the arrival of single cell transcriptomics, which have uncovered myofibroblastic

[1]Division of Molecular Pathology, Oncode Institute, The Netherlands Cancer Institute, Amsterdam, The Netherlands. [2]Division of Molecular Carcinogenesis, The Netherlands Cancer Institute, Amsterdam, The Netherlands. [3]Cancer Research UK Edinburgh Centre, Institute of Genetics and Cancer, University of Edinburgh, Edinburgh, UK. [4]Genomics Core Facility, The Netherlands Cancer Institute, Amsterdam, The Netherlands. [5]Flow Cytometry Facility, The Netherlands Cancer Institute, Amsterdam, The Netherlands. [6]Experimental Animal Pathology Facility, The Netherlands Cancer Institute, Amsterdam, The Netherlands. [7]Laboratory for Intravital Imaging and Dynamics of Tumor Progression, VIB Center for Cancer Biology, KU Leuven, Leuven, Belgium. [8]Department of Oncology, KU Leuven, Leuven, Belgium. ✉e-mail: j.houthuijzen@nki.nl; j.jonkers@nki.nl

or extracellular matrix (ECM) producing CAFs, antigen-presenting CAFs and inflammatory CAFs (iCAFs)[5–8]. Despite increasing knowledge regarding the functions and heterogeneity of CAFs, little is known about the origin of CAFs and whether the different sources of fibroblasts explain the observed heterogeneity. A number of hypotheses have been postulated regarding the cell-of-origin of CAFs, including recruitment of bone marrow (BM) or adipose tissue-derived (AT) mesenchymal stem cells (BM-MSCs or AT-MSCs), epithelial-to-mesenchymal transition (EMT) of tumor cells, trans-differentiation of endothelial cells or pericytes and recruitment and activation of local, tissue-resident fibroblasts[2]. Most of these hypotheses have been tested in co-culture settings or in studies involving co-injections of CAF precursors with tumor cells in mice[9–15] (11 and 12 are reviews). However, assessing the ability of established tumor cells to induce CAF-like behavior in a recipient cell in culture does not mean that these two cell types meet, co-exist or communicate with each other in a similar way during de novo tumorigenesis in vivo. Only few studies have examined the origin of fibroblasts in more physiological settings. Through BM transplantations and fibroblasts lineage tracing models LeBleu and colleagues showed that fibroblasts present in fibrotic kidney disease consisted of locally activated fibroblasts, BM-MSCs and trans-differentiated endothelium-derived fibroblasts[16]. In addition, BM transplantations revealed the existence of a population of BM-derived PDGFRα-negative CAFs in the MMTV-PyMT mouse model of breast cancer[17].

Single-cell transcriptomics datasets of pan-tissue fibroblasts[18] and studies of fibroblasts in normal human breast tissue[19] revealed two main subsets of fibroblasts present in all tissues with distinct gene expression profiles and CD26 expression. CD26 (also known as Dpp4) is a T-cell co-stimulatory molecule with peptidase activity and is involved in the deactivation of multiple hormones and cytokines[20–22] (20 and 22 are reviews). CD26+ fibroblasts have been linked to scar formation and fibrosis-related diseases[23–26].

Here, we show through a series of complementary transplantation techniques using genetically engineered mouse models (GEMMs) of invasive lobular breast cancer (ILC) and triple-negative breast cancer (TNBC), that the heterogeneous population of CAFs in these tumors originate almost entirely from local, tissue-resident normal fibroblasts (NFs). Assessment of early, progressive and end-stage tumors shows involvement of local, tissue-resident NFs at all stages of tumor development. Single-cell transcriptomics of mouse ILC lesions during tumor development reveal the simultaneous disappearance of NFs and appearance of CAFs. Using functional co-culture assays and in vivo analyses, we show that CD26+ NFs are predisposed to become iCAFs, whereas CD26− NFs give rise to myofibroblastic CAFs (myCAFs). CD26+ NFs enhanced the invasive properties of tumor cells via matrix metalloproteinase (MMP) activity and co-cultures of CD26+ NFs and tumor cells recruited CD11b+ myeloid cells in a CXCL12-dependent manner. Taken together, our data show that tissue-resident mammary fibroblasts contribute to the heterogeneous population of CAFs in mouse models of breast cancer and that CD26+ NFs are at the origin of pro-tumorigenic iCAFs.

## Results

### TNBC and ILC contain CAFs at all stages of tumor development but differ in their relative contents

To investigate and compare the contribution of CAFs to the TME of different breast cancer subtypes we used our established GEMMs that closely mimic human breast cancer histology of BRCA1-deficient TNBC and E-cadherin-deficient ILC. BRCA1-deficient TNBC can be modeled in vivo by Cre-conditional, mammary gland-specific loss of BRCA1 and P53 (*WapCre;Brca1^{F/F};Trp53^{F/F}*, in short WB1P) alone or in combination with Myc overexpression (*WapCre;Brca1^{F/F};Trp53^{F/F};Col1a1^{invCAG-Myc-IRES-Luc}*, in short WB1P-Myc)[27,28]. ILC was likewise modeled by Cre-mediated mammary-specific loss of E-cadherin and PTEN (*WapCre;Cdh1^{F/F};Pten^{F/F}*,

in short WEPtn), or by E-cadherin loss with mutant PIK3CA^{H1047R} overexpression (*WapCre;Cdh1^{F/F};Col1a1^{invCAG-Pik3caH1047R-IRES-Luc}*, in short WEH1047R)[29]. Histopathological analysis of the resulting mammary tumors showed that TNBC and ILC differed greatly in their CAF content. Sections of end-stage tumors from each of the four breast cancer mouse models and normal mammary glands were stained for fibroblasts using antibodies against PDGFRβ, αSMA and vimentin; for collagen fibers by Masson Trichrome and for epithelial cells using antibodies against E-cadherin and EpCAM (Fig. 1A). ILC lesions contained considerably more fibroblasts than TNBC lesions. Furthermore, the analysis of early, advanced and end-stage tumors isolated from the four mouse models showed a distinct, but remarkably stable presence of fibroblasts at all stages of tumor development (Fig. 1B). The observed differences in CAF content might be a consequence of tumor-specific interactions between CAFs and tumor cells or differences in the cells-of-origin of CAFs. Since little is known about the cells-of-origin of CAFs during de novo tumorigenesis in vivo, we set out to determine which precursor cells give rise to CAFs in these ILC and TNBC mouse models.

### Mammary tissue-resident cells contribute to pool of CAFs in mouse models of breast cancer

We designed a series of complementary transplantation studies to investigate whether CAFs in ILC and TNBC originate from tumor cells that underwent EMT, recruitment of BM-derived MSCs and/or recruitment of local, tissue-resident fibroblasts (Fig. 2A). Transplantation of pre-neoplastic mouse mammary epithelial cells (MMECs) derived from our ILC and TNBC mouse models into the cleared fat pads of *mTmG* (membrane-tdTomato-membrane-eGFP) recipient mice with ubiquitous expression of cell membrane-localized tdTomato resulted in de novo mammary tumor formation.

Flow cytometry analysis of early, advanced and end-stage tumors showed that nearly all CAFs (EpCAM−/CD45−/CD31−/PDGFRβ+ cells in WEPtn and WEH1047R tumors and EpCAM−/CD45−/CD31−/CD49f−/PDGFRβ+ cells in WB1P and WB1P-Myc tumors, see Supplementary Figs. 1 and 2 for gating strategy) expressed tdTomato and were therefore host-derived (Fig. 2B). This does not exclude the possibility that EMT did not occur in these tumors, as all tumors, independent of tumor stage, contained on average 11.08% (±5.5%) tdTomato-negative cells that lacked expression of epithelial markers (Supplementary Fig. 3A, B). Interestingly, these cells did not express the fibroblast-associated marker PDGFRβ (Supplementary Fig. 3A). Isolation and culture of EpCAM-positive tumor cells, EpCAM-negative tumor cells and tdTomato-positive CAFs derived from the same WB1P-Myc tumors showed that both EpCAM-positive and EpCAM-negative tumor cells lacked the ability to remodel collagen and gave rise to mammary tumors upon orthotopic re-transplantation. In contrast, tdTomato-positive CAFs were able to remodel collagen but did not lead to tumor formation in vivo (Supplementary Fig. 3C–E). Reports by Sarrio et al. and Bartoschek et al. have shown that the basal cell marker CD49f can be expressed by CAFs[30,31]. Since we used CD49f together with EpCAM to discriminate tumor cells in our WB1P and WB1P-Myc models, we potentially ignored tumor-cell derived CAFs in our analysis. Analysis of MMEC transplanted WB1P and WB1P-Myc tumors revealed that EpCAM−/CD49f+ cells accounted for 1.8% and 1.7% of non-endothelial and non-immune cells in WB1P and WB1P-Myc tumors, respectively (Supplementary Fig. 3F). Isolation and culturing of EpCAM−/CD49f+ cells from WB1P tumors showed that these cells were unable to contract collagen, suggesting that these cells do not function as bonafide CAFs (Supplementary Fig. 3G). However, these cells are bonafide tumor cells as they have lost the tumor suppressor genes *Trp53* and *Brca1* (Supplementary Fig. 3H).

Previous reports have shown that BM-MSCs can contribute to the pool of CAFs in breast cancer[13,15,17,32]. However, most studies have used xenograft models or co-injections of tumor cells with cultured BM-

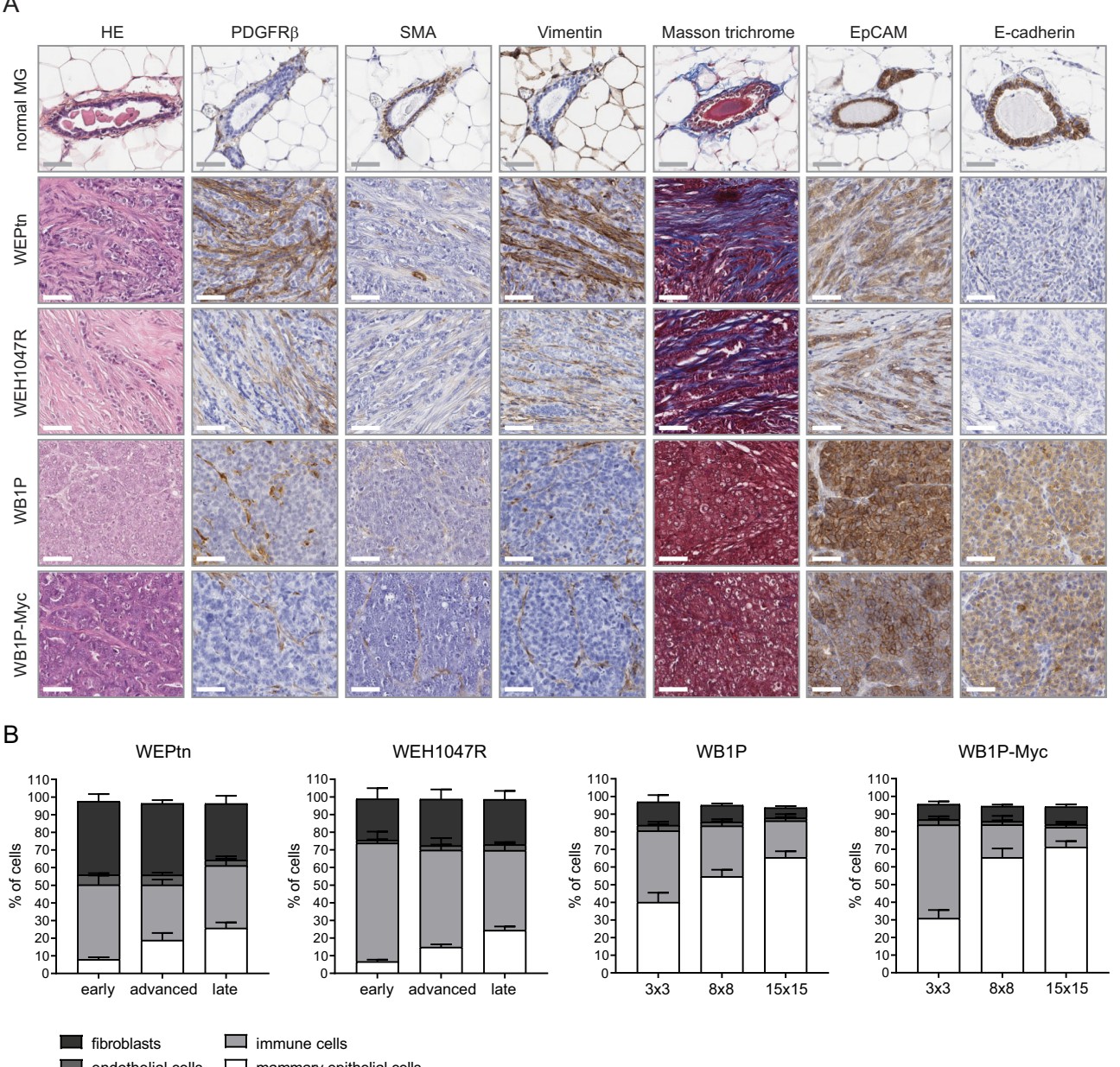

**Fig. 1 | TNBC and ILC differ in histology and quantity of CAFs. A** HE and immunohistochemical stainings of non-tumor bearing control mammary glands and tumors derived from ILC mouse models (WEPtn and WEH1047R) and TNBC mouse models (WB1P and WB1P-Myc) for fibroblast markers (platelet-derived growth factor receptor beta: PDGFRβ, alpha smooth muscle actin: SMA, vimentin), collagen (Masson trichrome) and epithelial cells (epithelial cell adhesion molecule: EpCAM and E-cadherin). Images are representatives of $n = 4$ mice per tumor model. Scale bars are 50 um. **B** Flow cytometry analysis of tumor composition at indicated stages of tumor development (WEPtn and WEH1047R) or tumor dimensions in mm

(WB1P and WB1P-Myc mice), minimum of $n = 6$ mice per time point. Mean percentage for every population is shown with error bars reflecting the standard error of the mean (SEM). Mammary epithelial cells (tumor cells) were defined as EpCAM+/CD49f+/CD31−/CD45−/PDGFRβ−. Endothelial cells were defined as CD31+/EpCAM−/CD45−. Immune cells were defined as CD45+/CD31−/EpCAM−. Fibroblasts in the WEPtn and WEH1047R models were defined as EpCAM−/CD45−/CD31−/PDGFRβ+. Fibroblasts in the WB1P and WB1P-Myc models were defined as EpCAM−/CD49f−/CD45−/CD31−/PDGFRβ+. Source data are provided in source data file.

MSCs. Here, we performed BM transplantations in our breast cancer mouse models to assess the contribution of BM-MSCs to the pool of CAFs. By using BM derived from *mTmG* donor mice, successful engraftment and BM-derived CAFs could be monitored by their tdTomato expression. Three weeks after BM engraftment of $Cdh1^{F/F};Pten^{F/F}$ (EPtn), $Cdh1^{F/F};Col1a1^{invCAG-Pik3caH1047R-IRES-Luc}$ (EH1047R), $Brca1^{F/F};Trp53^{F/F}$ (B1P) or $Brca1^{F/F};Trp53^{F/F};Col1a1^{invCAG-Myc-IRES-Luc/+}$ (B1P-Myc) female mice, tumors were induced by intraductal delivery of Cre-encoding lentivirus. Analysis of the resulting mammary tumors showed that nearly all CAFs in the TNBC and ILC models were tdTomato-negative (Fig. 2C).

Some tdTomato-positive CAFs were observed in early lesions (EPtn: 3.6% ± 4.1. EH1047R: 4.6% ± 2.1. B1P: 2.8% ± 3.6. B1P-Myc: 0.5% ± 0.6), but these numbers were not statistically significant, suggesting that BM-MSCs were not actively recruited during mammary tumorigenesis. BM harvested from tumor-bearing mice confirmed successful engraftment of donor BM and tdTomato expression in BM-MSCs (Supplementary Fig. 4).

Since these findings contradict previous reports based on the PyMT breast cancer mouse model[17], we questioned whether BM-MSC-derived CAFs might have lost tdTomato expression during

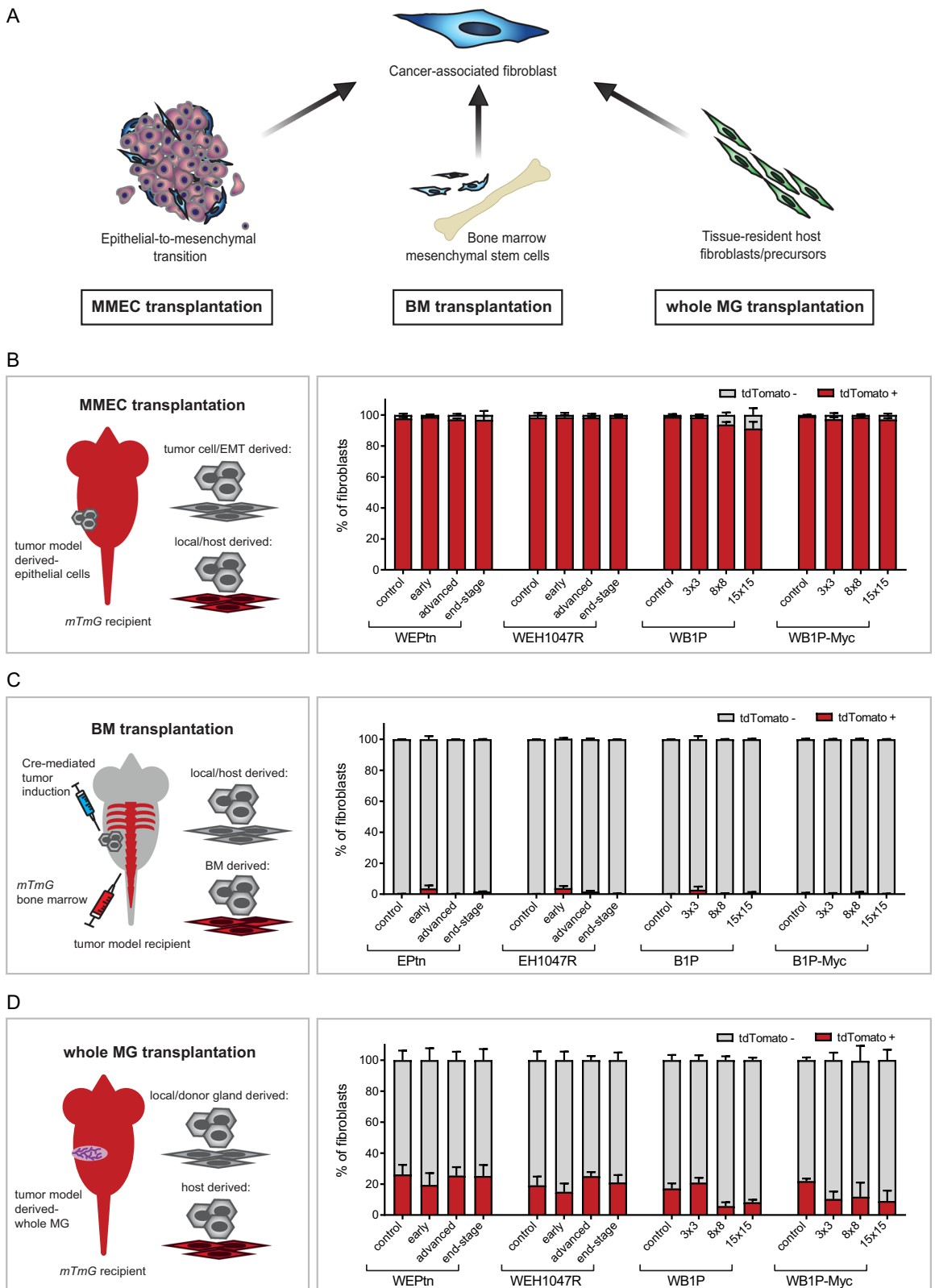

their recruitment and differentiation and repeated this experiment using *Cdh1^{F/F};Pten^{F/F};mTmG* mice transplanted with BM from wild-type FVB/n mice (Supplementary Fig. 5A). Now, the transplanted BM lacked fluorescence and the recipient mouse expressed the *mTmG* Cre-conditional reporter that replaced tdTomato expression with GFP in tumor cells (Supplementary Fig. 5B, C). Again, BM-MSCs were not the main contributors to the population of CAFs, as the majority of the

PDGFRβ+ and PDGFRβ− CAFs, in the resulting mammary tumors were tdTomato-positive (Supplementary Fig. 5D−F) and therefore derived from the host rather than the transplanted BM (Supplementary Fig. 5G).

To determine whether CAFs originate from tissue-resident fibroblasts or their precursors, we performed whole mammary gland transplantations[33]. Transplanting whole pre-neoplastic mammary

**Fig. 2 | Nearly all CAFs in TNBC and ILC originate from mammary tissue-resident fibroblasts. A** Schematic overview of most common CAF-cell-of-origin hypotheses and the transplantation approaches used to validate them. **B** Schematic representation of mouse mammary epithelial cell (MMEC) transplantation approach and the results of transplanting pre-neoplastic mammary fragments of WEPtn, WEH1047R, WB1P or WB1P-Myc into the cleared mammary fat pads of tdTomato-positive (mTmG) recipient mice. Tumors were analyzed at early (3 × 3 mm), advanced (8 × 8 mm) or end-stage (15 × 15 mm) by flow cytometry to determine the fraction of tdTomato+ and tdTomato− fibroblasts. Mice transplanted with littermate control, non-neoplastic tissue were used as controls. **C** Schematic representation of bone marrow transplantation experiment. tdTomato-positive bone marrow from mTmG mice was transplanted in lethally irradiated mice. Three weeks post transplantation, tumors were induced via intraductal injection of lentivirus expressing Cre-recombinase. Resulting tumors

were analyzed at indicated time points by flow cytometry to determine the fraction of tdTomato+ and tdTomato− fibroblasts. Mice that were intraductally injected with PBS and therefore did not develop tumors were used as controls. **D** Schematic representation of whole mammary gland transplantation and its results. Pre-neoplastic and littermate control 3rd mammary glands were harvested from WEPtn, WEH1047R, WB1P and WB1P-Myc mice and transplanted in mTmG mice. Tumors and controls were harvested at indicated time points and analyzed by flow cytometry to determine the fraction of tdTomato+ and tdTomato− fibroblasts. All experiments in **B**–**D** were performed with a minimum of 5 mice per time point and in these experiments fibroblasts were defined as EpCAM−/CD45−/CD31−/PDGFRβ+ (WEPtn and WEH1047R models) or EpCAM−/CD49f−/CD45−/CD31−/PDGFRβ+ (WB1P and WB1P-Myc models). Mean percentage for every population is shown with SEM. Source data for **B**–**D** are provided in source data file.

glands including the nipple and associated skin isolated from WEPtn, WEH1047R, WB1P and WB1P-Myc mice into *mTmG* recipients allowed for discrimination between gland- and host-derived cells. Analysis of tumors arising in the transplanted mammary glands revealed that the majority of CAFs were tdTomato-negative and therefore descendants of the transplanted gland rather than the host (Fig. 2D). Although a substantial number of CAFs were tdTomato-positive, this was also observed in recipients transplanted with control mammary glands from WapCre-negative littermates, indicating a surgery-induced effect (Fig. 2D). Nevertheless, the organization of the transplanted glands resembled endogenous gland architecture and cellular distribution (Supplementary Fig. 6). In summary, these transplantation-based in vivo studies showed that CAFs in TNBC and ILC largely originate from tissue-resident fibroblasts or precursors.

### Single-cell transcriptomics reveal NF and CAF dynamics during tumor development

Recent advances in single-cell transcriptomics have highlighted CAF heterogeneity and the existence of distinct CAF subpopulations in multiple cancer types[5,31,34]. Having shown that CAFs originate from mammary tissue-resident cells, we decided to investigate the heterogeneity of normal mammary fibroblasts (NFs) and their transition into CAFs during de novo tumorigenesis. For this purpose, we focused on ILC as these tumors show a large infiltrate of CAFs and little is known about their role in ILC development and progression. ILC mammary lesions were induced by intraductal injection of Cre-encoding lentivirus in EPtn mice and CAFs were isolated from these tissues 6-, 12- or 18-weeks post-injection. Together with control NFs, these CAFs were subjected to single-cell transcriptomics, which revealed a progressive disappearance of NF clusters and a concomitant appearance of CAF clusters during tumor progression each with distinct gene expression profiles (Fig. 3A–C). Gene ontology analysis of the significantly differentially expressed genes that define the CAF clusters in a one vs. rest comparison (Log2FC > 1.5, FDR < 0.05) identified specific functions for each CAF cluster such as immune-modulation (cluster 2, iCAFs) and ECM production and maintenance (cluster 4, myCAFs) (Fig. 3D, E and Supplementary Data 1). CAF cluster-specific gene expression signatures were generated by a one-vs.-rest cluster comparison with a log2FC > 1.5, FDR < 0.05 and a Rank-score > 20 to select for highly expressed genes. Additionally, the genes that met these criteria were filtered in a direct comparison of iCAF vs. myCAF, (or vice versa) for genes with a log2FC > 1.5, FDR < 0.05 and Rank-score > 20. This resulted in an iCAF gene signature of 14 genes and a myCAF gene signature of 53 genes (see Supplementary Data 1 and Supplementary Fig. 7A). Analysis of general fibroblast markers showed that only *S100a4* (Fsp1) and *Acta2* (smooth muscle actin, SMA) were differentially expressed between iCAFs and myCAFs (Supplementary Fig. 7B). Visualization of Fsp1 and SMA-positive fibroblasts in WEPtn, WEH1047R, WB1P and WB1P-Myc end-stage tumors showed that these cells indeed localized to the tumor stroma (Supplementary Fig. 7C). Using our single-cell

transcriptomics dataset we explored cell surface markers that could discriminate between iCAFs and myCAFs, to allow for sorting of live, primary CAFs from our GEMMs. CD34 was differentially expressed between iCAFs (CD34+) and myCAFs (CD34−) (Fig. 3F). Isolation of CD34+ and CD34− CAFs from WEPtn ILCs and subsequent qPCR or western blot analysis of iCAF and myCAF markers deducted from our signatures confirmed expression of iCAF markers in CD34+ CAFs and expression of myCAF markers in CD34- CAFs (Fig. 3H, I and Supplementary Fig. 7D). Furthermore, WEPtn-derived myCAFs showed superior collagen contractibility compared to iCAFs (Fig. 3J).

### iCAF and myCAF gene signatures are present in human breast CAFs

Using our mouse ILC-derived iCAF and myCAF gene signatures, we set out to determine if these gene signatures are specific to CAFs when applied to other datasets. First, we performed single-cell transcriptomics of end-stage tumors derived from WEPtn and WEH1047R mice to generate a dataset reflecting the entire TME of mouse ILC (Fig. 4A–C). Assessment of the gene signatures showed iCAF and myCAF gene expression specifically in cluster 3, which was defined as the fibroblast cluster based on *Col1a1* expression (Fig. 4D, E). Additionally, single-cell transcriptomics of WB1P and WB1P-Myc end-stage tumors also revealed iCAF and myCAF gene expression amongst fibroblast cluster 5 in these TNBC models (Fig. 4F–J). Reclustering of fibroblast cluster 3 from the WEPtn/WEH1047R dataset and fibroblast cluster 5 from the WB1P/WB1P-Myc dataset and removal of *Rgs5*-positive pericytes from analysis showed the presence of iCAFs and myCAFs in both ILC and TNBC mouse models (Supplementary Fig. 8A–F). In summary, our iCAF and myCAF gene signatures specifically mark CAFs in the context of the entire TME of both ILC and TNBC mouse models.

Recently, Gómez-Cuadrado et al. investigated the characteristics of the non-immune stroma in human ILC and IDC by laser-capture microdissection and transcriptomic analysis[35]. We interrogated this dataset to determine whether mouse iCAF and myCAF gene signatures were also expressed in human ILC and IDC. Various iCAF- and myCAF-related genes were expressed in human ILC and IDC stroma (Supplementary Fig. 8G, H). Expression of *CPXM2*, *CCN5*, *HMCN2*, *EMB*, *CAV1* and *CFB* could not be determined since these genes were not present in the IDC dataset. Single sample gene set enrichment analysis showed that the iCAF and myCAF signature scores were significantly enriched in tumor stroma compared to tumor epithelium, indicating that similar iCAF and myCAF populations exist in human ILC (Fig. 4K) and IDC (Fig. 4L).

Additionally, we interrogated single-cell transcriptomics data of 34 human breast cancers published by Pal et al.[36]. This dataset, comprised of ER-positive, PR-positive, HER2-positive and TNBC, showed a distinct clustering (Fig. 4M–O), with multiple *EPCAM*+ and *Keratin-8*+ (*KRT8*) clusters corresponding to the heterogeneous tumor cell populations in each of the breast cancer subtypes (Supplementary

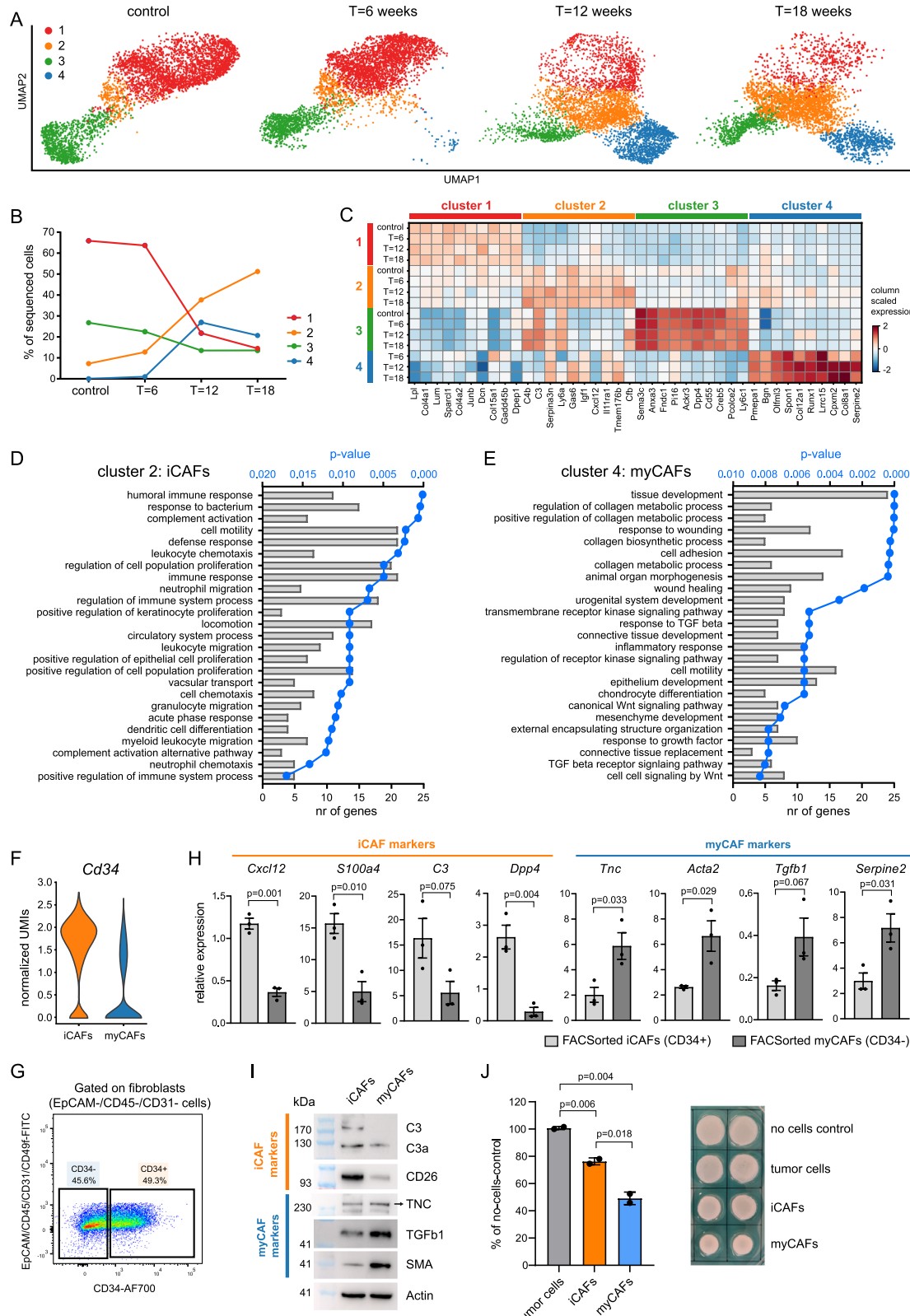

Fig. 8I). Clusters negative for *EPCAM* and *KRT8*, corresponded to the cells of the TME, like endothelial cells (*CD31*+), immune cells (*CD45*+) and fibroblasts (*COL1A1*+) and were present in all breast cancer subtypes (Fig. 4O and Supplementary Fig. 8J). Analysis of our signatures within this dataset showed that iCAF and myCAF gene signatures localized to the fibroblast cluster (defined as *COL1A1*-positive cells, Fig. 4P). Reclustering of the *COL1A1*-positive CAFs and removal of

*RGS5*-positive pericytes resulted in four CAF subpopulations, of which the two major clusters (cluster 1 and 2) expressed iCAF and myCAF gene signatures, respectively (Fig. 4Q), confirming the presence of iCAFs and myCAFs in human breast cancer.

Similar CAF subpopulations have been described in human pancreatic ductal adenocarcinoma (PDAC) and PDAC mouse models[5]. Interestingly, these PDAC iCAF and myCAF signatures showed

**Fig. 3 | Single-cell transcriptomics reveal dynamics in NFs and CAFs during tumor development. A** Fibroblasts were isolated from control, non-tumor bearing mice and ILC-tumor bearing mice (Cdh1[F/F];Pten[F/F] mice) 6, 12 or 18 weeks after intraductal lenti-Cre injection to initiate tumor formation (n = 2 mice per time point). Fibroblasts were defined as EpCAM−/CD45−/CD31− cells, sorted by FACS and subjected to single-cell RNA sequencing using the 10X Genomics platform. The data were analyzed using scanpy and represented as UMAP plots for each time point. **B** Dynamics of the four fibroblast clusters during tumor development. Number of fibroblasts in each cluster is expressed as percentage of all sequenced cells of the indicated time points. **C** Matrix plot showing the top-10 genes that define each cluster based on Rank-score and their expression at each time point. Gene ontology analysis of genes that define CAF cluster 2 (immune-modulating CAFs, iCAFs, **D**) and CAF cluster 4 (myofibroblastic CAFs, myCAFs, **E**). Top-25 significantly upregulated biological processes are shown in **D** and **E**. Full analysis can be found in Supplementary Data 1. Statistical significance was determined using Fisher's exact test with multiple comparison correction according to Benjamini–Hochberg. **F** Cd34 expression in iCAFs and myCAFs within single-cell transcriptomics dataset. **G** CD34 expression in CAFs within WEPtn-derived ILCs. Fibroblasts were defined as EpCAM−/CD45−/CD31− cells. **H** CD34− and CD34+ CAFs were isolated by FACS from WEPtn-derived tumors (n = 3 mice) and analyzed for iCAF and myCAF markers by qPCR. Gapdh was used as housekeeping gene. Data represents mean ± SEM. **I** Western blot analysis of primary iCAFs (CD34+) and myCAFs (CD34−) for iCAF and myCAF markers (n = 1). Actin was used as a loading control. **J** Three-day collagen contraction assay with primary iCAFs (CD34+), myCAFs (CD34−) or tumor cells (EpCAM+) (n = 2 biological replicates). Level of contraction was determined compared to no-cells control and plotted in graph on the left (mean ± SEM). Representative image shown on the right. Statistical significance in **H** and **J** was determined by two-tailed Student's t test. Source data for **H**–**J** are provided in source data file.

substantial overlap with the iCAF and myCAF clusters in ILC (Supplementary Fig. 9A–D). We did not find a distinct ILC CAF cluster that resembled the antigen-presenting CAFs (apCAFs) observed in PDAC, although some cells in the ILC iCAF cluster appeared positive for the PDAC apCAF signature (Supplementary Fig. 9A). Comparison of gene expression profiles from PDAC iCAFs and myCAFs with ILC iCAFs and myCAFs beyond the signatures showed significant correlations in both up- and downregulated genes in iCAFs and myCAFs from ILC and PDAC (Supplementary Fig. 9E, F), indicating that, independent of cancer type, these CAF subtypes emerge during tumor development.

**The mammary gland harbors two subtypes of normal fibroblasts**

The single-cell transcriptomics dataset of mammary NFs transitioning into CAFs during ILC tumorigenesis revealed two subpopulations of NFs in non-tumor bearing mammary glands (Fig. 3A). These NFs could be separated into CD26- (cluster 1) and CD26+ (cluster 3) cells (Fig. 5A). The presence of two distinct fibroblast populations that could be distinguished from each other by CD26 expression has previously been reported in normal human and mouse mammary tissue[19,37]. Gene ontology analysis of the differentially expressed genes that define CD26− and CD26+ NFs primarily revealed functions related to ECM production and maintenance (Supplementary Fig. 10A, B and Supplementary Data 1). Further investigation of the genes annotated to the common GO term "Extracellular matrix" revealed differences between CD26− and CD26+ NFs (Supplementary Fig. 10C). CD26− NFs primarily express collagens (Col15a1, Col18a1, Col4a1, Col4a2 and Col5a3) and proteins involved in collagen fibril formation, which have been previously linked to inhibition of angiogenesis, metastasis and tumor suppression (Dcn, Lum, Sparcl1 and Anxa5)[38–40]. In contrast, CD26+ NFs predominantly express fibronectin (Fn1), fibrillin (Fbn1) and elastic fiber-related ECM molecules (Emilin2) that have been associated with enhanced angiogenesis and metastasis (Cd248, Emilin2, Dpt)[41–44].

The presence of CD26− and CD26+ NFs in the mammary gland was confirmed by whole mount analysis and flow cytometry (Fig. 5B, C). Interestingly, a decrease in CD26+ fibroblasts was observed within the stroma of mammary tumors compared to wild type mammary glands, confirming the absence of CD26 in myCAFs and CD26 expression in iCAFs within the single-cell transcriptomics dataset (Fig. 5A, D). This shift in the ratio of CD26− and CD26+ fibroblasts was observed in both ILCs and TNBCs and became more apparent as tumors progressed (Fig. 5D and Supplementary Fig. 11). To determine if CD26− and CD26+ NFs contribute to specific CAF populations, we performed a trajectory analysis, which showed that both CD26− NFs and CD26+ NFs can transition into iCAFs and subsequently into myCAFs. In addition, a small fraction of CD26− NFs seemed to directly transition into myCAFs (Fig. 5E). These data suggest that both CD26− and CD26+ NFs contribute to iCAF and myCAF populations in mammary tumors. Another method to infer potential functional relations between clusters is by clustree analysis[45]. Clustree analysis shows a clear and early separation

of myCAFs from the other fibroblast clusters regardless of the resolution chosen to analyze the data (Fig. 5F). The iCAF cluster and CD26+ NF cluster appear to be related and display cluster instability and cross-over at higher resolutions. The CD26- NF cluster, like the myCAFs, appears to be more separate from other clusters (Fig. 5F). Both trajectory and clustree analysis suggest functional relations between CD26+ NFs and iCAFs, however the origin of myCAFs remains unclear from these analyses.

**CD26+ NFs enhance tumor cell invasion**

Next, we set out to investigate whether functional differences exist between CD26− and CD26+ NFs. Primary mammary CD26− and CD26+ NFs were harvested and cultured to determine whether they differentially affected tumor cell behavior. Both NF populations displayed spindle-shaped cell morphology and parallel alignment associated with cultured fibroblasts (Supplementary Fig. 12A). In a transwell setting, CD26+ NFs displayed preferential recruitment toward ILC-derived tumor cells, whereas TNBC cells recruited both CD26− and CD26+ NFs (Fig. 6A–C). Interestingly, when CD26− and CD26+ NFs were mixed prior to plating, both NF populations migrated toward ILC tumor cells, suggesting crosstalk between CD26− and CD26+ NFs in this experimental setting (Supplementary Fig. 12B, C). CAFs have previously been shown to enhance the metastatic potential of tumor cells[2,46–51]. To determine whether CD26− or CD26+ NFs specifically promoted tumor cell migration, they were subjected to an organotypic invasion assay to measure migration of tumor cells into a matrix containing collagen and basement membrane extract (BME) with or without NFs. Although both CD26− and CD26+ NFs promoted tumor cell invasion, migration into the matrix was enhanced by CD26+ NFs compared to CD26− NFs. This effect was most prominent when CD26+ NFs were combined with ILC-derived tumor cells (Fig. 6D, E), but was also observed with TNBC-derived tumor cells (Supplementary Fig. 12D, E). Together, these findings suggest that CD26+ NFs are at the origin of pro-tumorigenic CAFs.

**CD26− and CD26+ NFs are predisposed to become myCAFs and iCAFs, respectively**

Next, we determined if both CD26− and CD26+ NFs were able to adopt an iCAF-like gene signature, as suggested by the trajectory analysis. For this purpose, we cultured primary CD26− and CD26+ NFs in conditioned medium (CM) derived from WEPtn tumor cells and compared their gene expression profiles with those of NFs cultured in control medium. Hierarchical clustering of the top-100 most variable expressed genes showed that CD26− and CD26+ NFs responded differently to the same tumor CM (Supplementary Fig. 13A). KEGG pathway analysis showed enrichment in pathways associated with cytokine-cytokine receptor interaction and chemokine signaling in CD26+ NFs cultured in CM compared to control CD26+ NFs. CD26− NFs exposed to tumor CM showed enrichment in pathways associated with tight junction regulation and cellular contractility (Supplementary Fig. 13B, C). The

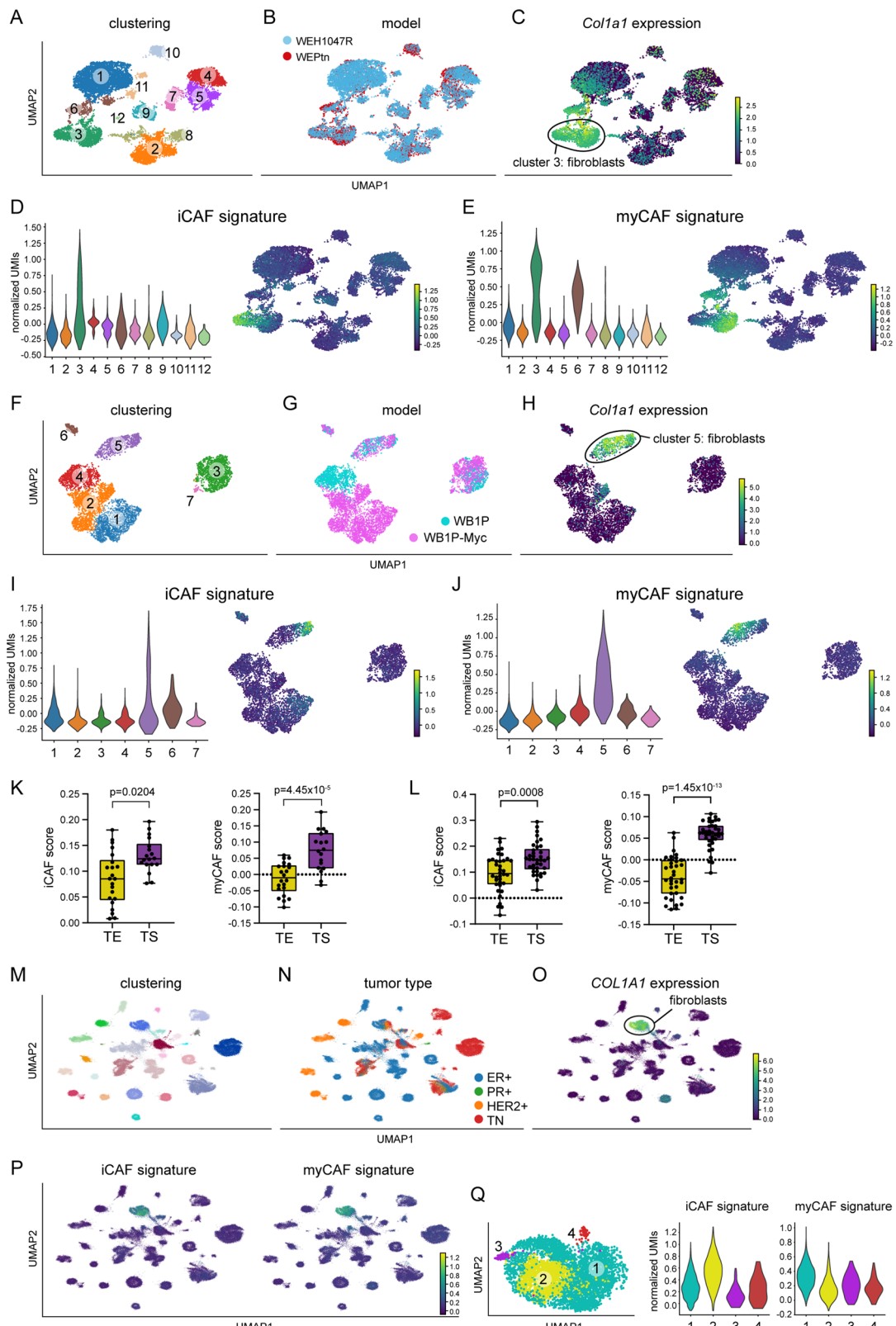

KEGG pathway analysis results founded our hypothesis that CD26+ NFs contribute to the population of iCAFs and could indicate that CD26− NFs contribute to the population of myCAFs. To investigate this in more detail, we investigated whether the genes of the iCAF and myCAF signatures from our single-cell transcriptomics dataset were upregulated or enriched in CM-exposed CD26+ and CD26− NFs. The iCAF gene signature was enriched in the CD26+ NFs cultured in tumor CM

(Supplementary Fig. 14A, B, D), but several genes of the iCAF signature were already expressed by CD26+ NFs compared to CD26− NFs in control conditions. The myCAF signature was not enriched in CD26− or CD26+ NFs cultured in tumor CM (Supplementary Fig. 14C, D). Since this experiment was performed in the absence of ECM and direct tumor cell-fibroblast contact, we repeated the experiment by co-culturing tumor cells and NFs in a collagen-rich matrix

**Fig. 4 | iCAFs and myCAFs are present in mouse models of breast cancer and human breast cancer.** Single-cell transcriptomics of WEPtn ($n = 3$ mice) and WEH1047R ($n = 3$ mice) tumors showing clustering (**A**), distribution of clusters amongst models (**B**) and *Col1a1* expression (**C**). **D** iCAF gene signature expression in WEPtn and WEH1047R. Data represented as violin plot and UMAP. **E** myCAF gene signature expression in WEPtn and WEH1047R. Data represented as violin plot and UMAP. Single-cell transcriptomics of WB1P and WB1P-Myc tumors ($n = 2$ mice per model) showing clustering (**F**), distribution of clusters amongst models (**G**) and *Col1a1* expression (**H**). **I** iCAF gene expression signature in WB1P and WB1P-Myc dataset. Data shown as violin plot and UMAP. **J** myCAF gene expression signature in WB1P and WB1P-Myc dataset. Data shown as violin plot and UMAP. **K** Single sample gene set enrichment analysis of iCAF and myCAFs scores in laser-microdissected human ILC samples ($n = 17$) separated in tumor epithelium (TE) and tumor stroma (TS). **L** Single sample gene set enrichment analysis of iCAF and myCAFs scores in laser-microdissected human IDC samples ($n = 36$) separated into tumor epithelium (TE) and tumor stroma (TS). Boxplots in **K** and **L** show whiskers ranging from minimum to maximum and, the box reflecting the 25th–75th percentile, with a line at the mean. Significance in **K** and **L** was determined using paired two-tailed Student's *t* test. Single-cell transcriptomics analysis of 34 human breast cancers showing clustering (**M**), tumor types (ER+, PR+, HER2+ and TNBC, **N**) and *COL1A1* expression (**O**). **P** iCAF and myCAF gene expression signatures on entire dataset. **Q** Reclustering of fibroblast cluster (COL1A1-positive cluster, without RGS5 + pericytes) and expression of iCAF and myCAF signature as violin plots.

(Supplementary Fig. 14E). Analysis of the CAF signatures within this experimental set-up showed an absence of the iCAF signature in both CD26− and CD26+ NFs co-cultured with tumor cells (Supplementary Fig. 14F, H) and a minor, yet not statistically significant difference in the myCAF signature (Supplementary Fig. 14G, H). However, the myCAF signature was enriched in CD26− NFs co-cultured with tumor cells compared to CD26+ NFs co-cultured with tumor cells. The induction of iCAF and myCAF phenotypes was not achieved robustly in culture and moreover highly dependent on the culture conditions used. To determine if CD26− and CD26+ NFs in culture still represent their in vivo counterparts we analyzed CD26 expression by flow cytometry in freshly isolated CD26− and CD26+ NFs (referred to as primary NFs) and CD26− and CD26+ NFs that were cultured for 1 week. Notably, CD26 expression rapidly decreased in cultured CD26+ NFs (Supplementary Fig. 14I, J). These results show that cultured fibroblasts diverge substantially from their in vivo counterparts, even though they retain some features such as differential response to tumor-derived CM and induction of tumor cell invasion. Hence, caution is needed when drawing conclusions about fibroblast functionality based solely on in vitro experiments. Therefore, we investigated whether a potential predisposition of NFs to iCAFs and myCAFs also existed in vivo. Previous reports showed that Engrailed1 (En1) marks a pro-fibrotic lineage of fibroblasts in the skin and that En1-positive skin fibroblasts also express CD26[26]. Therefore, *En1-Cre;mTmG* mice would allow for in vivo tracing of GFP-positive CD26+ NFs and tdTomato-positive CD26− NFs during tumor development independent of changes in gene expression. For this purpose we performed MMEC transplantations in *En1-Cre;mTmG* mice (Fig. 7A). However, contrary to the expected results, in the mammary gland not all En1+ NFs expressed CD26 and a substantial amount of CD26+ NFs do not originate from the En1+ lineage (Fig. 7B, C). MMEC transplantations were performed as described previously[52,53], with the exception of clearing of the recipient mammary fat pad (Fig. 7A). By keeping the recipient gland intact and placing donor tissue at the dorsal side of the fourth mammary gland, we ensured minimal perturbations to the recipient gland and the fibroblasts present in that gland. Transplantation of control donor tissue from WapCre-negative *Cdh1^{F/F};Pten^{F/F}* mice in *En1-Cre;mTmG* mice verified proper outgrowth and comparable distributions of fibroblasts in transplanted and control glands (Fig. 7B–D). Analysis of the mammary tumors that arose from transplantations of pre-neoplastic mammary donor tissue from WEPtn, WB1P and WB1P-Myc mice in *En1-Cre;mTmG* mice revealed an increase in En1+/CD26− (En1+) and EN1−/CD26− (double-) fibroblasts, a decrease in EN1−/CD26+ (CD26+) fibroblasts and similar levels of EN1+/CD26+ (double+) fibroblasts in tumors compared to controls (Fig. 7E–G). Transcriptomics analysis and hierarchical clustering of the top-100 most variable genes within En1+, CD26+, double+ and double− fibroblasts isolated from tumors and control mammary glands showed that clustering did not depend on En1 status, but rather on CD26 expression, as double− and En1+ fibroblasts clustered together and CD26+ and double+ fibroblasts clustered together (Fig. 7H). For simplicity we collectively refer to En1+ and double− fibroblasts as CD26− fibroblasts and CD26+ and double+ fibroblasts as CD26+ fibroblasts. Analysis of the myCAF and iCAF signatures within this dataset showed that the iCAF signature was enhanced in CD26+ CAFs, whereas the myCAF signature was predominantly expressed by CD26− CAFs (Fig. 7I, J). Single sample gene set enrichment analysis verified that CD26+ CAFs significantly upregulated the iCAF gene signature compared to CD26− CAFs. Both CD26− and CD26+ CAFs upregulated myCAF-associated genes compared to CD26− and CD26+ NFs, but CD26− CAFs were superior to CD26+ CAFs in their myCAF gene expression (Fig. 7K). In line with these findings both primary CD26− and CD26+ NFs were able to contract collagen in the presence of WEPtn tumor cells compared to NFs plated alone. A small increase in collagen contraction was observed for CD26− NFs co-cultured with tumor cells compared to CD26+ NFs co-cultured with tumor cells, however this effect was not statistically significant (Fig. 7L). Taken together, these results show that an in vivo predisposition exists for CD26+ NFs to become iCAFs that also harbor some myCAF functionality, whereas CD26- NFs become myCAFs and do not contribute to the population of iCAFs.

## CD26+ NFs co-cultured with tumor cells secrete CXCL12 to recruit monocytes

Our single-cell transcriptomics showed that iCAFs express several cytokines (*Ccl2, Ccl7, Ccl8, Cxcl2* and *Cxcl12*) that play a role in the recruitment of myeloid cells (Fig. 8A, B). Cytokine array analysis of CM derived from CD26− and CD26+ NFs co-cultured with tumor cells revealed that the co-culture of CD26+ NFs and tumor cells released more CXCL2 and CXCL12 than the co-culture of CD26− NFs and tumor cells. Conversely, co-cultures of CD26− NFs and tumor cells secreted more TNFα and CD54 compared to co-cultures of CD26+ NFs and tumor cells (Fig. 8C and Supplementary Fig. 15A, B). Secretion of CXCL12 from fibroblasts has previously been linked with enhanced tumor growth, angiogenesis and recruitment of T-regulatory cells[34,54]. To determine if CD26+ NFs and their release of cytokines upon co-culture with tumor cells are involved in the recruitment of immune cells, we harvested splenocytes and monitored their recruitment in transwell assays toward CM derived from NF mono- or co-cultures with tumor cells. We found that CM of CD26+ NFs co-cultured with tumor cells recruited more CD11b+ monocytes than all other mono- and co-cultures, indicating an immune-modulatory function of CD26+ NFs (Fig. 8D). No differences were observed in the ability of the NF mono- or co-cultures to recruit CD3+ T-cells or B220+ B-cells (Supplementary Fig. 15C, D). Neutralizing antibodies against CXCL2 and CXCL12 revealed that only inhibition of CXCL12 completely abrogated the recruitment of monocytes (Fig. 8D). Since CXCL12 has also been associated with fibroblast-induced tumor cell migration[47,49,55], we assessed whether CD26+ NF-derived CXCL12 was also responsible for the observed invasion of tumor cells in the organotypic invasion assays (Fig. 6D). Addition of CXCL12 neutralizing antibodies did not affect CD26+ NF-induced tumor cell invasion (Fig. 8E, F). In addition, MMPs have been shown to play an important role in tumor cell migration and

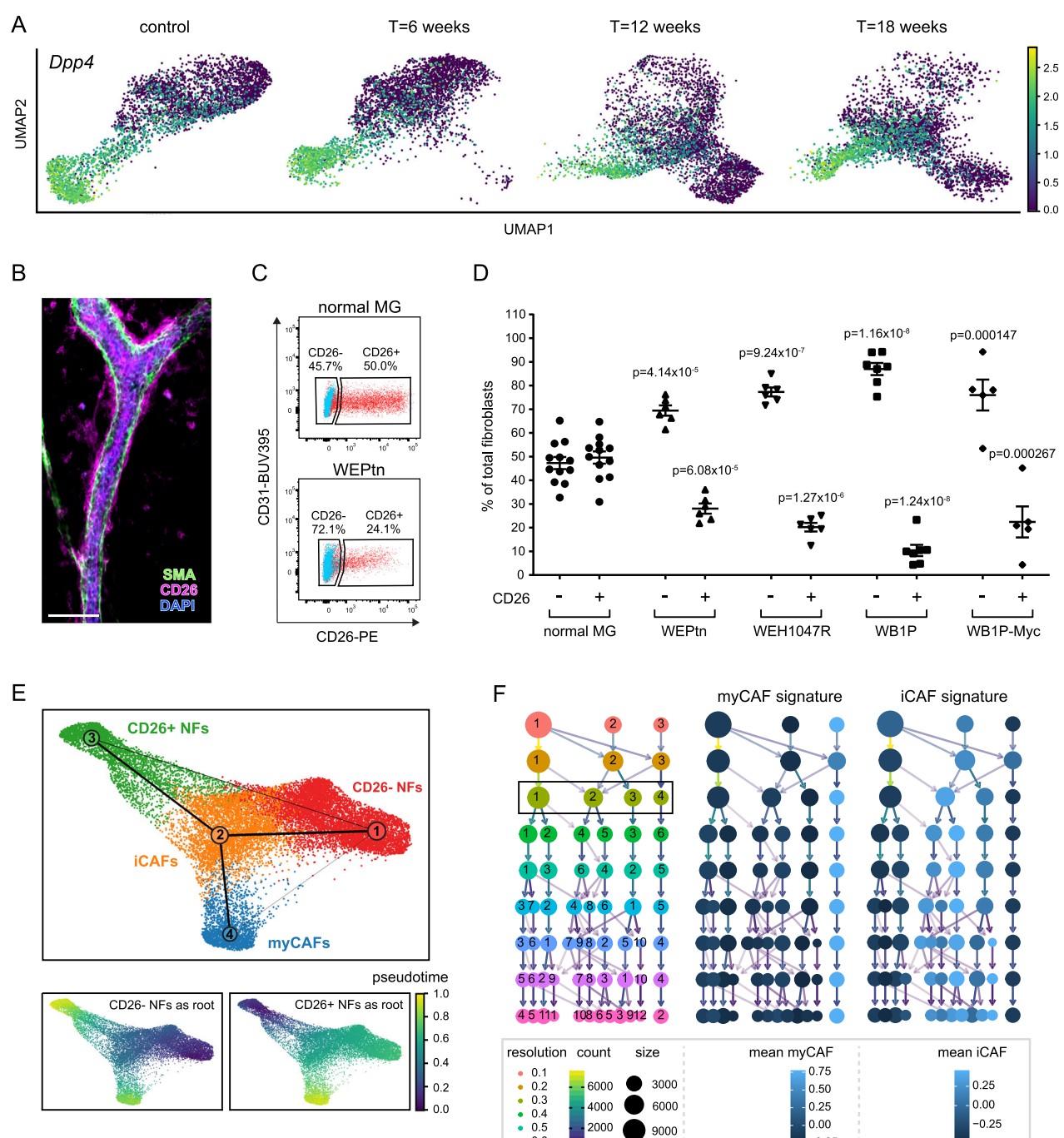

**Fig. 5 | Ratio of CD26− and CD26+ fibroblasts change during tumor development and both CD26− and CD26+ NFs contribute to CAF populations. A** UMAP plot of single-cell transcriptomics data of NFs and CAFs isolated during mammary tumorigenesis in WEPtn mice (Fig. 3A) showing *CD26* (*Dpp4*) expression. **B** Whole mount analysis of normal mammary gland stained for CD26 (in magenta) and smooth muscle actin (SMA, in green) and dapi (in blue). Scale bar is 50 um. Representative image is shown from two biological replicates with similar results. **C** Flow cytometry analysis of normal mammary gland and WEPtn-derived ILC gated on fibroblasts (EpCAM−/CD45−/CD31− cells) shows the presence of CD26− and CD26+ fibroblasts. Shown in blue is fluorescence-minus-one (FMO) control and shown in red is full stained sample. FMO control was used to set gates. **D** Percentage of CD26− and CD26+ fibroblasts in control mammary glands (*n* = 12 mice) and end-stage tumors derived from WEPtn (*n* = 6 mice), WEH1047R (*n* = 6 mice), WB1P (*n* = 7 mice) and WB1P-Myc mice (*n* = 5 mice). A two-way ANOVA with Bonferonni multiple

comparison correction was used to determine statistical significance by comparing CD26− or CD26+ NFs of control to the CD26− or CD26+ CAFs of the different mouse models. Individual data points reflect biological replicates (mice) with mean ± SEM. Source data are provided in source data file. **E** Partition-based graph abstraction (PAGA) trajectory analysis of single-cell transcriptomics data from Fig. 3A indicates trajectory of NF clusters toward CAF clusters. Pseudotime was plotted using CD26− NFs as a starting point (root) and CD26+ NFs as a starting point. **F** Clustree analysis of single-cell transcriptomics of fibroblasts during tumorigenesis at various resolutions to show functional relationships between clusters. Box indicates resolution of 0.3 used to analyze the data resulting in 4 clusters: CD26− NFs (cluster 1), iCAFs (cluster 2), CD26+ NFs (cluster 3) and myCAFs (cluster 4). Middle panel and right panel show myCAF and iCAF gene expression signature plotted on clustree, respectively. Light blue indicates high iCAF/myCAF score, dark blue indicates low score.

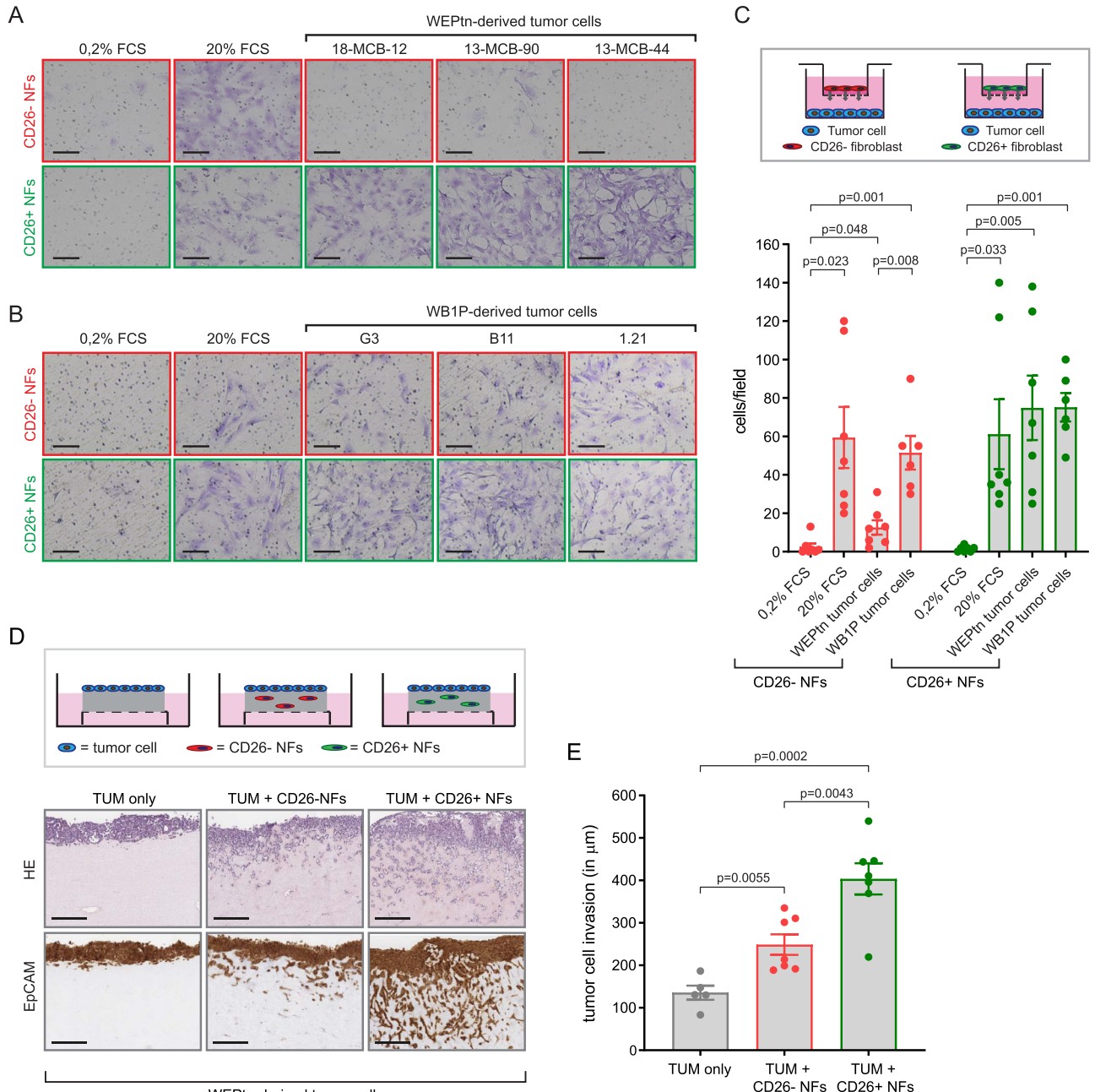

**Fig. 6 | CD26+ NFs are recruited toward tumor cells and induce invasiveness.**
**A**, **B** Representative result of fibroblast migration toward tumor cells. Transwell assays were used to assess the recruitment of CD26− and CD26+ fibroblasts toward ILC-derived tumor cells (**A**) and TNBC-derived tumor cells (**B**). Tumor cells were plated in the bottom compartment and fibroblasts were plated in Matrigel-coated inserts. 24 h after plating the bottom-side of the inserts were fixed and stained with crystal violet to visualize the migrated cells. All assays were done in low serum conditions (0.2% FCS). 20% FCS and 0.2% FCS alone were used as positive and negative controls, respectively. Scale bars are 120 um. **C** Quantification of transwell assays ($n = 7$ independent experiments with similar outcome, WB1P $n = 6$). Statistical significance was determined using two-tailed Student's $t$ test. Bars represent mean ± SEM. **D** Representative result of organotypic invasion assay, in which collagen-containing gels are loaded with CD26− or CD26+ NFs or left empty. Tumor cells are plated on top of these gels and invasion into the gels is assessed after 1 week. Gels are processed as HE slides or stained for EpCAM to visualized tumor cells. Scale bars are 200 um. **E** Quantification of organotypic invasion assays based on EpCAM staining ($n = 5$ for TUM only, $n = 7$ for TUM + NFs conditions). All replicates shown are independent experiments with similar outcome. Invasion was measured in um from top of the gel to the invasive front of the tumor cells. Bars represent mean ± SEM. Statistical significance was determined using two-tailed Student's $t$ test. Source data for **C** and **E** are provided in source data file.

invasion[56]. Inhibition of MMPs using marimastat, a broad spectrum MMP inhibitor, blocked tumor cell invasion induced by CD26− and CD26+ NFs (Fig. 8E, F). In line with these results, we found significant changes in gene expression of MMPs in CD26− and CD26+ NFs cultured in tumor CM compared to NFs cultured in control medium. When comparing differential MMP expression in CD26− NFs cultured in tumor CM with CD26+ NFs cultured in tumor CM, we found significant changes in the expression of MMP1, 3, 9, 11, 12, 13 and 28. Two of these MMPs, MMP1 and 9, are drug targets of marimastat, and are significantly increased in CD26+ NFs compared to CD26− NFs (Fig. 8G). Taken together, our results showed that nearly all CAFs in our breast cancer models originate from tissue-resident mammary fibroblasts and that CD26− NFs are predisposed to become myCAFs, whereas CD26+ NFs are predisposed to become pro-tumorigenic

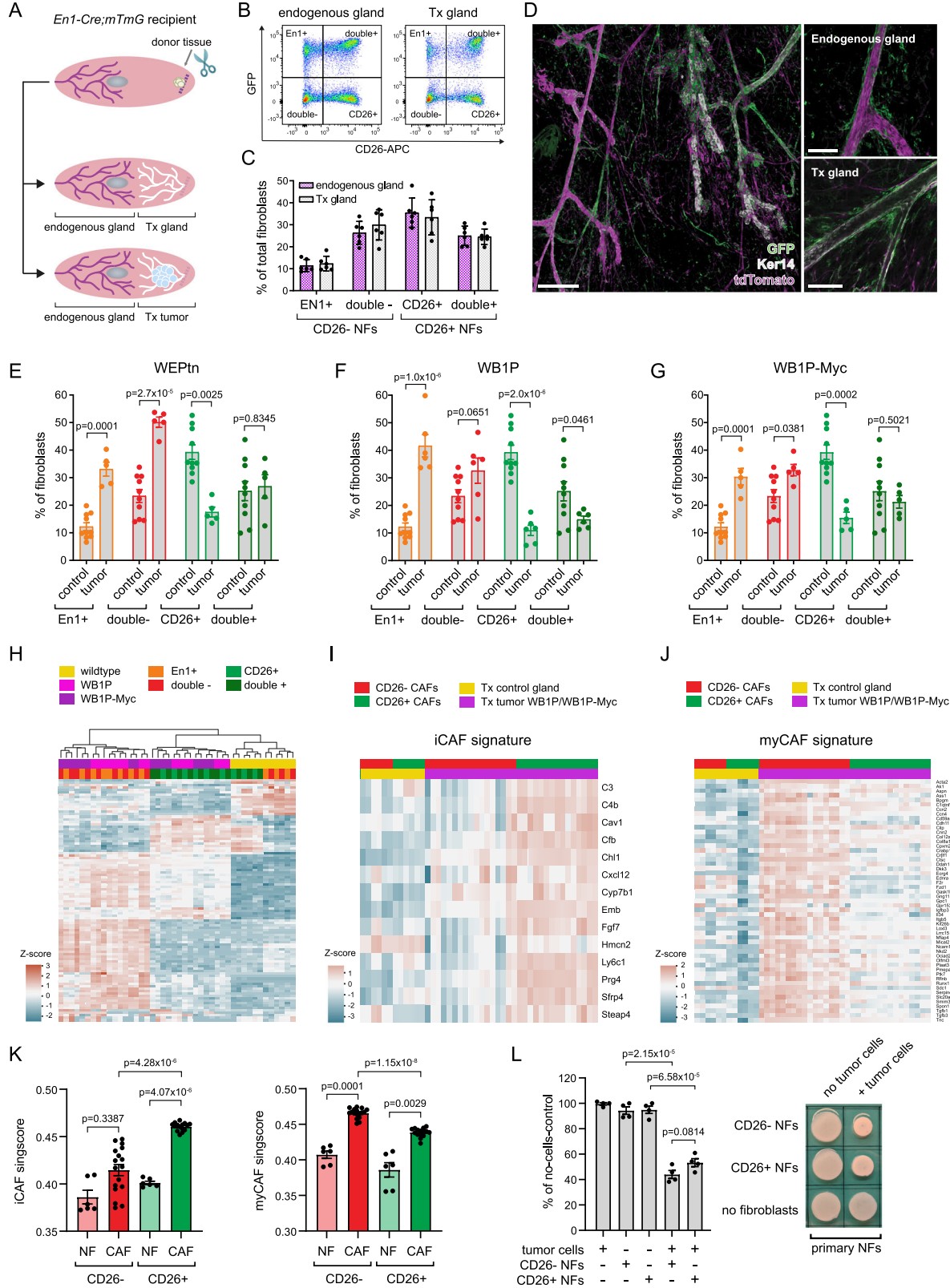

iCAFs, capable of recruiting myeloid cells via CXCL12 secretion and enhancing tumor cell invasion via MMP signaling (Fig. 8H).

## Discussion

Here, we have shown through a series of complementary transplantation techniques using multiple GEMMs, that nearly all CAFs in breast cancer originate from tissue-resident fibroblasts, with little to no contribution from EMT or BM precursors. In our transplantation studies we made use of PDGFRβ to mark the fibroblasts, as this was the most widely expressed fibroblast-associated marker in both CAFs and NFs our models. However, others have shown differences in PDGFRβ expression in tumor settings[57]. In our mouse models, PDGFRβ reliably discriminated EMT tumor cells from bonafide CAFs, but this may not be the same in other models. The lack of BM involvement in our mouse

**Fig. 7 | CD26− NFs are predisposed to become myCAFs, whereas CD26+ NFs become iCAFs. A** Schematic representation of MMEC transplantations in *En1-Cre;mTmG* mice. **B** Representative flow cytometry plots of the distribution of mammary fibroblasts in *En1-Cre;mTmG* control mice and *En1-Cre;mTmG* mice transplanted with control mammary tissue. Fibroblasts were defined as EpCAM −/CD49f−/CD31−/CD45− cells. **C** Quantification of indicated fibroblasts populations in endogenous mammary glands of *En1-Cre;mTmG* mice and mammary glands with transplanted control donor tissue (n = 6 mice per group). Statistical significance was determined using two-tailed Student's *t* test. No statistical differences found between control and transplanted (Tx) glands for the different populations. Data shows as mean ± SEM. **D** Whole mount analysis of transplanted gland showing the endogenous mammary ductal structure in magenta (tdTomato) and the outgrown transplanted donor tissue in white (stained for keratin 14). Fibroblasts from the Engrailed1 lineage are shown in green (GFP). Zoom images show GFP+ fibroblasts surrounding both endogenous and transplanted mammary ducts. Scale bar of overview image is 200 um. Scale bar in image of endogenous gland is 50 um and scale bare in image of Tx gland is 100 um. Representative image is shown from two biological replicates with similar results. Analysis of fibroblast distribution in control transplanted mice (n = 10) and mice transplanted with pre-neoplastic WEPtn donor tissue (n = 5, **E**), WB1P donor tissue (n = 6, **F**) or WB1P-Myc donor tissue (n = 5, **G**). All tumors were analyzed at end-stage (WEPtn: 42-weeks post transplantation, WB1P and WB1P-Myc tumor size: 15 × 15 mm). Data shown as mean ± SEM. Statistical

significance of **E**–**G** was determine by unpaired Student's *t* test comparing control to tumor for each of the populations. **H** Transcriptomics analysis and hierarchical clustering of En1+, CD26+, double+ and double− fibroblasts isolated from control transplanted and WB1P or WB1P-Myc transplanted En1-Cre;mTmG mice. Top 100 most variable genes are shown here. Fibroblasts were isolated by FACS and defined as EpCAM−/CD49f−/CD31−/CD45−. iCAF (**I**) and myCAF (**J**) gene signatures of CD26− and CD26+ NFs and CAFs isolated from *En1-Cre;mTmG* mice transplanted with control or pre-neoplastic mammary tissue from WB1P and WB1P-Myc mice. Fibroblasts were isolated by FACS from control transplanted mice (n = 3) and end-stage tumors (WB1P n = 5, WB1P-Myc n = 4). **K** Single sample gene set enrichment analysis (Singscore) of the iCAF and myCAF signatures in CD26− and CD26+ NFs (n = 3 mice, 2 samples per mouse for CD26− NFs (En1+ and double−) and 2 samples per mouse for CD26+ NFs (CD26+ and double+)) and CAFs (n = 9 mice (WB1P and WB1P-Myc combined), 2 samples per CAF population) isolated from *En1-Cre;mTmG* mice. Statistical significance was determine by two-tailed Student's *t* test. **L** Quantification and representative image of 3-day collagen contraction assay with primary CD26− and CD26+ NFs in the presence or absence of WEPtn-derived tumor cells. Level of contraction was determined compared to no-cells control. Results shown are from four independent experiments with similar outcome. Statistical significance was determine using two-tailed Student's *t* test. Data in **K** and **L** are mean ± SEM. Source data for **C**, **E**–**G**, **K** and **L** are provided in source data file.

mammary tumor models was surprising, since Raz et al. showed in vivo recruitment of BM-MSCs in tumors isolated from MMTV-PyMT mice[17]. The differences between our results and those of Raz et al. may be due to experimental design. The rapid development of mammary tumors in the MMTV-PyMT model may create an overlap between the time of BM transplantation and tumor development which could influence recruitment, especially since irradiation can lead to a tissue damage response which BM-MSCs are likely to home to (reviewed in ref. [58]). In our experimental set-up BM engraftment preceded tumor induction by three weeks, reducing the change of interference between the two processes.

Single-cell transcriptomics of fibroblasts at several stages of tumor development have revealed a gradual shift of two NF subpopulations into two CAF populations with distinct functions. Throughout this study we used CD26 as a marker to separate the two NF populations identified by single-cell transcriptomics. This is consistent with previous reports of CD26+ and CD26− fibroblasts in human and mouse mammary tissue[19,37]. In normal human breast tissue, CD26+ NFs reside around ducts whereas CD26− NFs are located around lobules[19]. Mouse mammary tissue lacks this organizational distinction between ducts and lobules and rather consists of ducts and terminal end buds which can both harbor CD26+ NFs. A recent study by Buechler et al. generated a single-cell transcriptomics atlas of mouse fibroblasts isolated from various tissues. They identified two main populations (*Pi16*+ and *Col15a1*+) present in all investigated tissues[18]. Integration of this data with single-cell transcriptomics data of mammary fibroblasts by Yoshitake et al. showed that the *Pi16*+ fibroblasts overlapped with the mammary CD26+ fibroblasts described by Yoshitake et al.[37], and that CD26+ NFs harbor immune-regulatory functions. In addition, the CD26− NFs overlapped with *Col15a1*+ fibroblasts described by Buechler et al. Interestingly, Yoshitake et al. have found that mammary CD26− and CD26+ fibroblasts displayed population-specific responses to estrogen treatment, indicating different roles of these fibroblasts in mammary gland development and maintenance. These results support our observations of the differential responses of CD26− and CD26+ NFs to tumor CM and co-cultures with tumor cells. CD26 is a known co-stimulatory molecule involved in T-cell activation[20]. In addition, the extracellular domain of CD26 has enzymatic activity that plays an important role in the inactivation of signaling molecules including incretins, chemokines and cytokines[59].

iCAFs and myCAFs constitute the majority of CAFs found in both ILC and TNBC. Interestingly, these CAF populations showed significant overlap in gene expression with PDAC-derived iCAFs and myCAFs. Additionally, analysis of the previously published pancreatic iCAF and myCAF gene expression signatures[5] within our dataset, showed expression of iCAF genes in CD26+ NFs and myCAF genes in CD26− NFs, suggesting a relationship between CD26− NFs and myCAF and CD26+ NFs and iCAFs. Furthermore, potential functional relationship between CD26+ NFs and iCAFs was also observed using clustree and trajectory analysis. Others have identified iCAFs and myCAFs in a number of malignancies including pancreatic cancer, colorectal cancer, breast cancer and other solid tumors[5,34,60−62], suggesting that the emergence of these CAF subtypes is a pan-cancer effect. Surprisingly, the origin of this heterogeneity is understudied. In pancreatic cancer it has been shown that TGF-β and IL1 are tumor-secreted factors that aid in the transformation of pancreatic stellate cells toward a myCAF or iCAF phenotype, respectively[61]. In addition, Miyazaki et al. has shown that differential exposure to Wnt drives the transformation of AT-MSCs into iCAFs or myCAFs in culture settings[10]. However, these studies did not consider NF heterogeneity, making it unclear if the observed CAF heterogeneity is a direct consequence of pre-existing heterogeneity within the cells of origin in vitro and in vivo. Here, we report a predisposition of normal mammary fibroblast subtypes to develop into functionally distinct CAF populations. Our findings appear to contrast other studies describing phenotypic states of CAFs rather than defined lineages. Hutton et al. has shown that the pancreas harbors two subtypes of fibroblasts that can be distinguished by CD105 expression. In pancreatic cancer both CD105− and CD105+ can adopt an iCAF and myCAF phenotype, however only CD105− pancreatic fibroblasts displayed tumor suppressive properties, suggesting at least some predispositioning with regards to tumor-suppressive CAFs in PDAC[63]. Using various co-cultures systems, we uncovered that CD26+ NFs adopt an iCAF phenotype whereas CD26− NFs are more likely to adopt a myCAF phenotype. However, the observed CAF subtype was highly dependent on the culture system used and analysis of primary vs. cultured fibroblasts revealed rapid loss of CD26 marker expression already shortly after culturing these cells. These results emphasize that caution is needed when drawing conclusions based on in vitro assays using highly plastic cells such as fibroblasts and that in vivo validation is crucial. In vivo analysis of CD26− and CD26+ fibroblasts in *En1-Cre;mTmG* mice confirmed that CD26− fibroblasts are superior at

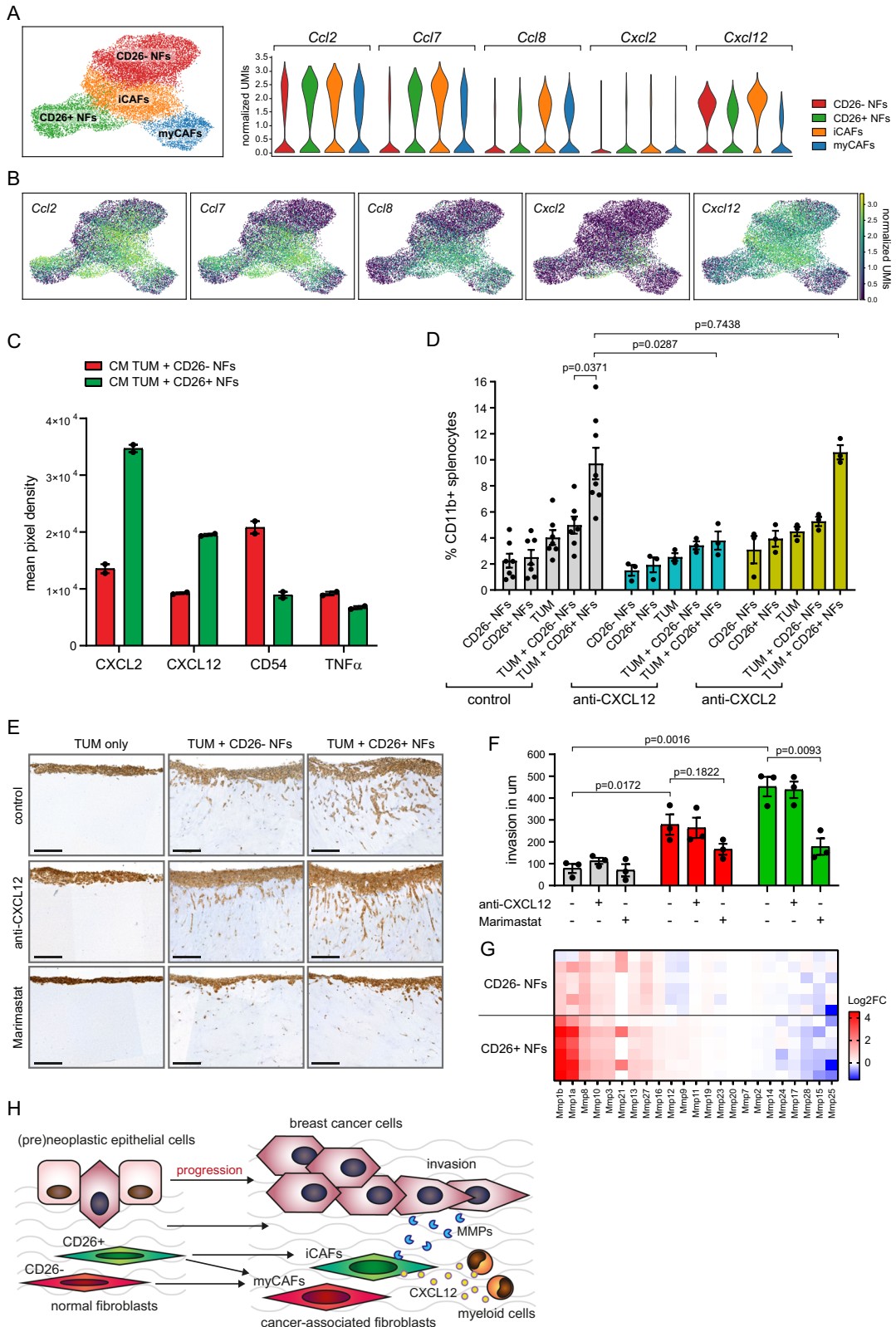

adopting a myCAF signature compared to CD26+ fibroblasts, whereas and CD26+ fibroblasts exclusively adopt an iCAF signature. These findings may appear to contradict other reports that ascribe a pro-fibrotic phenotype to CD26+ fibroblasts[23,26]. However, our in vivo analysis showed that multiple myCAF-related genes were also upre-gulated in CD26+ CAFs compared to CD26+ NFs, indicating that CD26+ NF-derived iCAFs also possess myofibroblastic functions which may

suggest a general activation state inherently related to fibroblasts or increased plasticity among CD26+ NFs compared to CD26− NFs. In contrast, iCAF-related genes were not expressed by CD26− CAFs. Although our functional assays and in vivo lineage tracing showed a specific predisposition of NFs toward iCAFs and myCAFs, the in silico trajectory analysis did not. The trajectory analysis predicted the transformation of both CD26− and CD26+ NFs into iCAFs first,

**Fig. 8 | CD26+ NFs recruit CD11b+ myeloid cells in a CXCL12 dependent manner and induce tumor cell invasion via MMPs. A** UMAP plot of single-cell transcriptomics dataset (all time points combined) showing the NF and CAF clusters and violin plots of the expression levels of *Ccl2, Ccl7, Ccl8, Cxcl2* and *Cxcl12*. **B** UMAP plot representation of data in **A. C** Up- and downregulated cytokines present in the conditioned medium of indicated co-cultures of tumor cells and fibroblasts as determined by cytokine array (*n* = 1 with two technical replicates). Data represented as mean ± SEM. **D** Results of transwell assays used to investigate recruitment of splenocytes toward the CM of fibroblast and tumor cell mono- or co-cultures in the presence or absence of CXCL2 or CXCL12 neutralizing antibodies. Migrated splenocytes were harvested from bottom compartment after 24 h, stained for CD11b, quantified by flow cytometry and displayed as percentage of CD11b+ splenocytes from total live single cells. Results shown are from *n* = 3 (anti-CXCL12 and anti-CXCL2 conditions) and *n* = 7 (control) independent experiments with similar outcome. Data represented as mean ± SEM. Statistical significance was determined using a two-tailed Student's *t* test. **E** Representative images of organotypic invasion

assays (EpCAM staining to visualize tumor cells) in the presence or absence of CXCL12 neutralizing antibodies or the MMP inhibitor Marimastat. Scale bar is 200 um. **F** Quantification of organotypic invasions assays in **E**. Gray bars represent data of TUM only, red bars represent data of TUM + CD26− NFs and green bars represent data of TUM + CD26+ NFs. Data shown as mean ± SEM. Invasion was measured in um from top of the gel to the invasive front of the tumor cells. Statistical significance was determine using two-tailed Student's *t* test. Results are shown from three independent experiments with similar outcome. **G** Log2 foldchange (Log2FC) in expression of MMPs in CD26− and CD26+ NFs cultured in tumor CM compared CD26− and CD26+ NFs cultured in control CM. **H** Schematic representation of the transition of CD26− and CD26+ NFs toward myCAFs and iCAFs, respectively during mammary tumorigenesis. CD26+ NFs transform into pro-tumorigenic iCAFs that release CXCL12 to recruit CD11b+ myeloid cells and induce tumor cell migration via MMPs. Source data for **C**, **D**, **F** and **G** are provided in source data file.

followed by subsequent transformation into myCAFs. The existence of multiple precursors that give rise to various cellular end-stages complicates these types of in silico analyses and may not accurately reflect the biological processes in vivo.

CD26+ NFs enhanced tumor cell invasion via MMP-activity and recruited monocytes in a CXCL12-dependent manner, indicating that the CAFs derived from CD26+ NFs have a pro-tumorigenic phenotype. Targeting these CAFs may be especially valuable in the context of ILC, since targeted therapies for this breast cancer subtype are limited and CAFs are abundantly present in these invasive tumors. Further investigation is needed to determine which tumor-secreted factors are important in driving iCAF and myCAF phenotypes and how inhibition of iCAF and myCAF functions may impact tumor progression and therapy response, with the ultimate goal of designing CAF-targeted therapies directed solely at the pro-tumorigenic functions of CAFs.

## Methods
### Mice
All animal experiments were approved by the Dutch Animal Ethical Committee and conducted in compliance with the Netherlands Cancer Institute and Dutch Animal Welfare guidelines. Generation of *Wap-Cre;Cdh1^{F/F};Pten^{F/F}* (WEPtn) mice has been described previously[29]. *WapCre;Cdh1^{F/F};Col1a1^{invCAG-Pik3caH1047R-IRES-Luc}* (WEH1047R) mice were generated by cloning human *Pik3ca* bearing the constitutively activating mutation H1047R in the *Frt-invCag-IRES-Luc* shuttle vector using FseI and PmeI, resulting in *Frt-invCag-Pik3ca^{H1047R}-IRES-Luc*. Flp-mediated knockin of the shuttle vector in *WapCre;Cdh1^{F/F}* GEMM-ESCs was performed as described previously[64]. Chimeric animals were crossed with *WapCre;Cdh1^{F/F}* mice to generate the experimental animal cohorts. Generation of *WapCre;Brca1^{F/F};P53^{F/F}* (WB1P) and *WapCre;Brca1^{F/F};Trp53^{F/F};Col1a1^{invCAG-Myc-IRES-Luc}* (WB1P-Myc) mice was previously described[28]. All breast cancer GEMMs are FVB/n background. *mTmG* reporter mice[65] were backcrossed for seven generations to FVB/n background to accommodate transplantations with donor tissue from our FVB/n-based breast cancer mouse models. *En1-Cre* mice were purchased from The Jackson Laboratory (JAX stock number:007916) and backcrossed with *mTmG* (FVB/n) mice for 2 generations to generate *En1-Cre;mTmG* mice for in vivo lineage tracing. All mice were housed on standard 12 h day/night cycle, in individually ventilated cages with ad libitum food. Room temperature was 21 °C with a humidity of 55%. All surgeries were performed under isoflurane anesthesia and carprofen pain medication. All animal experiments were conducted with female mice. The number, age and duration of each experiment is described in the main text or in the relevant figure legends. Maximum permitted cumulative tumor volume of 2000 mm³ was not reached in any of the experiments. All animals were euthanized using $CO_2$.

### Reagents
The following flow cytometry antibodies were used throughout this study: EpCAM-PE-Cy7 (Invitrogen, ebioscience clone G8.8), E-cadherin-PE-Cy7 (Biolegend clone DECMA-1), CD49f-AF700 (R&D systems clone GoH3), CD45-AF700 (Invitrogen, ebioscience clone 30-F11), CD45-BUV805 (BD biosciences clone 30-F11), CD45-FITC (Invitrogen, ebioscience clone 30-F11), CD31-BUV395 (BD biosciences clone 390), PDGFRβ-APC (Invitrogen, ebioscience clone APB5), CD26-APC (Biolegend clone H194-112), CD26-PE (Biolegend clone H194-112), CD3-BUV395 (BD biosciences clone 500A2), B220-PE-Cy7 (Invitrogen, ebioscience clone RA3-6B2), CD11b-APC (Invitrogen, ebioscience clone M1/70), CD45-PerCP (BD biosciences clone 30-F11), Sca1-APC-Cy7 (BD biosciences clone D7), CD90.1-FITC (Invitrogen, ebioscience clone HIS51). All flow cytometry antibodies were used in a 1:100 dilution, incubated for 30 min on ice. For the whole mount analysis the following antibodies were used: Keratin-14 (rabbit, Covance, PRB155P), alpha-Smooth muscle actin (mouse IgG2a, clone 1A4, Sigma-Aldrich), and CD26 (rabbit monoclonal, Abcam, clone EPR18215). Secondary antibodies: goat anti-rabbit or goat anti-mouse IgG2a conjugated to Alexa-647 (Thermo Fisher, A21245 and A21241 respectively) and donkey anti-rabbit conjugated to Alexa-568 (Thermo Fisher, A10042). The following antibodies were used for immunohistochemistry and immunofluorescence: PDGFRβ (Cell signaling, #3196), SMA (Thermo Scientific, RB-9010), vimentin (Cell signaling, #5741), EpCAM (Abcam, ab32392), E-cadherin (Cell signaling, #3195), rat-anti-Keratin 8 (TROMA-1, DSHB), rabbit-anti-Keratin 14 (Abcam, EPR17350), rabbit-anti-FSP1 (Abcam, EPR14639(2)) and mouse-anti-SMA (Sigma-Aldrich, clone 1A4). The following neutralizing antibodies were used: anti-CXCL12 (R&D systems, clone 79014, 100 ug/ml) and anti-CXCL2 (Thermo Fisher, clone 40605, 50 ug/ml). The following small molecule inhibitors were used: Marimastat (MMP inhibitor, BB-2516, Selleckchem, 100 nM).

### Cell lines
All cells were maintained at 37 °C and 5% $CO_2$. WEPtn tumor cell lines were derived from end-stage primary WEPtn tumors. TNBC cell lines were derived from end-stage primary WB1P tumors. Tumor samples were processed into a single cell suspension as described in "flow cytometry analysis" and cultured in DMEM/F12 medium containing 10% FCS, pen/strep (50 units/ml), EGF (5 ng/ml), insulin (5 ug/ml) and cholera toxin (5 ng/ml). Three rounds of differential trypsinization were used to remove fibroblasts from the culture. To ensure all cell lines were depleted from fibroblasts, they were analyzed for their EpCAM expression by flow cytometry. Only cell lines with more than 90% EpCAM+ cells and no expression of PDGFRβ were used for subsequent experiments. In addition, the cells were genotyped for the relevant alleles to ensure recombination of tumor driver genes (*Cdh1* and *Pten* or *Brca1* and *Trp53*).

## Immunohistochemistry

All immunohistochemical stainings were performed on FFPE material. Slides were deparafinized and rehydrated followed by antigen retrieval in Tris/EDTA (pH 9.0). Endogenous peroxidase was blocked using 3% $H_2O_2$. Next the slides were blocked using either 10% non-fat milk in PBS (for EpCAM, PDGFRβ and SMA stainings) or 4% BSA + 5% normal goat serum (NGS) in PBS (for E-cadherin and vimentin stainings). Incubation with primary antibodies was done overnight at 4 °C using the following dilutions: PDGFRβ (Cell signaling, #3196) 1:50, alpha-SMA (Fisher scientific, RB-9010) 1:200, vimentin (Cell signaling, #5741) 1:200, EpCAM (Abcam, ab32392) 1:200 or E-cadherin (Cell signaling, #3195) 1:200. All primary antibodies were diluted in 1% BSA + 1.25% NGS in PBS. EnVision+ HRP-conjugated anti-rabbit (ready-to-use, Dako Agilent, K400311-2) was used as secondary antibody followed by DAB/$H_2O_2$ development and counterstaining with hematoxylin.

## Immunofluorescent labeling and whole-mount imaging

Mammary glands were dissected and incubated in a mixture of collagenase I (1 mg/ml, Roche Diagnostics) and hyaluronidase (50 µg/ml, Sigma-Aldrich) at 37 °C for 20 min prior to fixation in periodate–lysine–paraformaldehyde buffer (1% paraformaldehyde (Electron Microscopy Science), 0.01 M sodium periodate, 0.075 M L-lysine and 0.0375 M P-buffer (0.081 M Na2HPO4 and 0.019 M NaH2PO4; pH 7.4) for 2 h at room temperature. Next, whole glands were incubated in blocking buffer containing 1% bovine serum albumin (Roche Diagnostics), 5% NGS (Monosan) and 0.8% Triton X-100 (Sigma-Aldrich) in PBS for at least 3 h at RT. Primary antibodies were diluted in blocking buffer and incubated overnight at room temperature whilst gently shaking. Secondary antibodies diluted in blocking buffer were incubated for at least 8 h. Nuclei were stained with DAPI (0.1 µg/ml; Sigma-Aldrich) in PBS. Glands were washed with PBS and mounted on a microscopy slide with Vectashield hard set (H-1400, Vector Laboratories). Primary antibodies: anti-K14 (rabbit, Covance, PRB155P, 1:700), anti-Smooth muscle actin (mouse IgG2a, clone 1A4, Sigma-Aldrich, 1:600), and anti-CD26 (rabbit monoclonal, Abcam, clone EPR18215, 1:200). Secondary antibodies: goat anti-rabbit or goat anti-mouse IgG2a conjugated to Alexa-647 (Thermo Fisher, A21245 and A21241 respectively, 1:400) and donkey anti-rabbit conjugated to Alexa-568 (Thermo Fisher, A10042, 1:400). Whole-mount mammary glands were imaged on an inverted Leica TCS SP8 confocal microscope, equipped with a 405 nm laser, an argon laser, a DPSS 561 nm laser and a HeNe 633 nm laser. Different fluorophores were excited as follows: DAPI at 405 nm, GFP at 488 nm, Tomato or Alexa-568 at 561 nm, and Alexa-647 at 633 nm. Images were acquired with a ×25 water immersion objective with a free working distance of 2.40 mm (HC FLUOTAR L ×25/0.95 W VISIR 0.17). Areas of interest were imaged using Z-stacks of 200 µm with an average Z-step size of 2 µm.

## Immunofluorescent staining of iCAFs and myCAFs

Formaldehyde-fixed paraffin-embedded sections from end-stage tumors of WEPtn, WEH1047R, WB1P and WB1P-Myc tumors were deparaffinated and rehydrated in a series of xylene and alcohol rehydrated. Antigen retrieval was performed by cooking the slides in Tris/EDTA buffer (pH 9.0). Tissues were blocked for 1 h in 1% non-fat milk, 5% NGS (Life Technologies) in TBST and incubated with primary antibodies: rat-anti-Keratin 8 (TROMA-1, DSHB) 1:200, rabbit-anti-Keratin 14 (Abcam, EPR17350) 1:1000, rabbit-anti-FSP1 (Abcam, EPR14639(2)) 1:2000 or mouse-anti-SMA (Sigma-Aldrich, clone 1A4) 1:400 in 5% goat serum, overnight at 4 °C. Serial stainings for FSP1 and Keratin-14 were performed on WB1P and WB1P-Myc tumors as both antibodies were produced in rabbits. The following secondary antibodies were used: goat-anti-mouse Alexa Fluor 488 (Thermo Fisher, A-11001), goat-anti-rabbit Alexa Fluor 568 (Thermo Fisher, A-11011), goat-anti-rat Alexa Fluor 647 (Thermo Fisher, A-21247) and goat-anti-rabbit Alexa Fluor 647 (Thermo Fisher, A-21244), all at a dilution of 1:1000 in 5% goat serum in TBST for 1 h at room temperature. Sections were embedded in vectashield hardset with dapi and imaged on a Leica TCS SP8 confocal microscope equipped with 6 lasers covering 9 laser lines (405 nm diode laser, 442 nm diode laser, 458, 476, 488 and 514 nm from argon laser, 561 nm diode laser, 594 nm HeNe laser and 633 nm HeNe laser). Areas of interest were imaged with 10 um Z-stacks (max projection) with an average Z-step of 0.4 um. Images were analyzed using Imaris software (version 9.8).

## Mouse mammary epithelial cell (MMEC) transplantation

Small tissue fragments of precancerous or control mammary glands were harvested from 4- to 6-week old donor mice and transplanted into the cleared 4th mammary fat pads of 3-week old *mTmG* recipients according to previously published protocols[52,53]. Transplantations in the *En1-Cre;mTmG* recipients were done without clearing of the recipients' fat pads as it is unclear at which stage and from which direction EN1+ fibroblasts populate the developing mammary gland. For these experiments the donor tissue was placed at the dorsal side of the 4th mammary gland to ensure minimal perturbations to the recipient gland and full potential of all fibroblasts present with the gland. Tumor outgrowth was monitored by palpation and mice were sacrificed at early, advanced and end-stage of tumor growth. Tumors and control tissues were harvested for flow cytometry analysis. For the ILC mouse models (WEPtn and WEH1047R) this required analysis at 18 (early), 24 (advanced) and 30 (end-stage) weeks after transplantation. Tumors from WB1P and WB1P-Myc mice were analyzed when they measured $3 \times 3$ mm (early), $8 \times 8$ mm (advanced) or $15 \times 15$ mm (end-stage).

## Bone marrow transplantation

$Cdh1^{F/F};Pten^{F/F}$ (EPtn), $Cdh1^{F/F};Col1a1^{invCAG-Pik3caH1047R-IRES-Luc}$ (EH1047R), $Brca1^{F/F};P53^{F/F}$ (B1P) and $Trp53^{F/F};Col1a1^{invCAG-Myc-IRES-Luc}$ (B1P-Myc) recipient mice were lethally irradiated with a single dose of 9 Gy at the age of 8 weeks. Age- and sex-matched *mTmG* donor mice were sacrificed using $CO_2$ and both femurs were isolated for BM harvesting. Femurs were flushed with DMEM + 5% FCS to extract the BM. The cells were filtered over 40 um cell strainers and spun down for 5 min at $300 \times g$. The cell pellets were resuspended in PBS and intravenously injected into irradiated mice. BM from one donor mouse was injected into one recipient mouse. Three weeks after BM engraftment the recipient mice were intraductally injected with lentivirus expressing Cre-recombinase to induce tumor formation. Tumor growth was monitored by palpation and mice were sacrificed at early, advanced and end-stage of tumor growth. For the ILC mouse models (EPtn and EH1047R) this required analysis at 6 (early), 12 (advanced) and 18 (end-stage) weeks after intraductal injection. Tumors from B1P and B1P-Myc mice were analyzed when they measured $3 \times 3$ mm (early), $8 \times 8$ mm (advanced) or $15 \times 15$ mm (end-stage).

## Whole mammary gland transplantation

Whole mammary gland transplantations were performed as described previously by Thompson et al.[33]. In brief, third mammary glands including nipple and surrounding skin of 4-week old donor mice were isolated and placed in cold PBS while the recipient mouse was prepared for surgery. Recipient mice of 4 weeks old were anaesthetized using isoflurane. A small circular patch of skin was removed between the 3rd and 4th mammary gland, in line with nipples of the recipient mouse. To accommodate the donor gland, the skin was separated from underlying abdominal wall, starting from the incision site dorsally toward the back. A suture was fastened at the dorsal tip of the donor gland and by using the suture the gland was guided in place in the space created along the flank of the recipient mouse. The tip of the donor gland was sutured to the dorsal skin of the mouse. The skin surrounding the nipple of the donor gland was sutured to the skin of

the recipient mouse, allowing the ventral side of the donor gland to be positioned on top of the blood vessel running from 3rd to 4th mammary gland. Tumor growth was monitored by palpation and tumors were analyzed at early, advanced and end-stage of tumor growth. For the ILC mouse models (WEPtn and WEH1047R) this required analysis at 12 (early), 18 (advanced) and 24 (end-stage) weeks after transplantation. Tumors from WB1P and WB1P-Myc mice were analyzed when they measured 3 × 3 mm (early), 8 × 8 mm (advanced) or 15 × 15 mm (end-stage).

## Flow cytometry analysis

All tumors and control mammary glands were placed in PBS on ice upon harvesting. Samples were chopped into small pieces using a scalpel and processed into a single cell suspension using a digestion mix containing 2 mg/ml collagenase + 4 ug/ml DNase in DMEM/F12. Samples were incubated for 60 min at 37 °C under continuous shaking. After incubation the collagenase was inactivated by addition of equal volume of DMEM + 5% FCS. Samples were filtered through 70 um cell strainers and spun at 300 g for 5 min to pellet the cells. Cell pellets were resuspended in red blood cell lysis buffer (RBC lysis, 155 mM NH4Cl, 10 mM KHCO3 and 0.1 mM EDTA in H2O) and incubated on ice for 5 min. Next the samples were spun down, 300 g for 5 min at 4 °C. Cell pellets were resuspended in FACS buffer (1% BSA + 5 mM EDTA in PBS) and stained with appropriate antibodies. For WEPtn and WEH1047R the samples were stained with CD45-AlexaFluor700, CD31-BUV395, EpCAM-PE-Cy7, CD26-APC or PDGFRβ-APC. For WB1P and WB1P-Myc the samples were stained with CD45-FITC, CD31-BUV395, EpCAM-PE-Cy7, E-cadherin-PE-Cy7, CD49f-AF700, PDGFRβ-APC or CD26-APC. Tumors transplanted in *En1-Cre;mTmG* mice were analyzed using the following antibodies: WEPtn and WEH1047R: EpCAM-PE-Cy7, CD31-BUV395, CD45-BUV805, CD26-APC. WB1P and WB1P-Myc: EpCAM-PE-Cy7, E-cadherin-PE-Cy7, CD49f-AF700, CD31-BUV395, CD45-BUV805. To assess successful engraftment of *mTmG* BM in the BM-transplanted mice we harvested femurs from the mice at time of sacrifice. The femurs were flushed with DMEM + 5% FCS and filtered through a 70 um cell strainer and spun down to pellet cells (300 g, 5 min). Red blood cells were removed by RBC lysis. Next the samples were spun down (300 g, 5 min, 4 °C) and resuspended in FACS buffer and stained with the following antibodies: CD45-PerCP, Ly6a-APC-Cy7 and CD90.1-FITC. All samples were run on the LSRII SORP flow cytometer with FACS DiVa software version 8.0.1 from Becton Dickinson, San Jose, CA, USA. FlowJo v10.8.0 software was used for analysis.

## Sorting and culture of primary fibroblasts

The 3rd, 4th and 5th mammary glands were harvested from female mice between 8 and 16 weeks of age. On average 4–6 mice were pooled for one sorting experiment. Samples were processed as described in the section "flow cytometry analysis". Samples were stained using the following antibodies: EpCAM-PE-Cy7, CD49f-PE-Cy7, CD45-AlexaFluor700, CD31-FITC, CD26-APC. Cells lacking expression of EpCAM, CD49f, CD45 and CD31 were considered fibroblasts. Sorting was done on a BD FACS Aria Fusion at 20 psi using a 100 um nozzle. CD26− and CD26+ fibroblasts were collected, spun down (300 g, 5 min) and plated in collagen type I-coated plates (8 ug/cm²) in DMEM + 20% FCS. All primary fibroblasts were cultured for no more than 6 passages. Experiments using primary fibroblasts were performed at the lowest possible passage numbers, typically passage 2 to 3.

## Sorting of iCAFs and myCAFs and qPCR analysis

ILC tumors were harvested from 20 week old WEPtn mice and processed into a single-cell suspension as described in the section "flow cytometry analysis". Samples were stained with the following antibodies: EpCAM-FITC, CD45-FITC, CD31-FITC, CD49f-FITC and CD34-AF700. EpCAM, CD45, CD31 and CD49f were used to gate out non-fibroblast cells. Cells negative for these markers were considered

fibroblasts and sorted based on CD34 expression (CD34-negative and CD34-positive). After sorting cells were either plated in DMEM + 20% FCS for in vitro experiments, or directly lysed in TRIsure (Bioline) for RNA extraction according to the manufacturers protocol. RNA was reverse transcribed using the Tetro cDNA synthesis kit from Bioline using oligodTs according to the manufacturers protocol with one alteration: incubation step at 45 °C was done for 60 min instead of 30 min. cDNA was used as input for qPCR of iCAF and myCAF markers using 2x Sensimix SYBRGreen low-ROX mastermix (Bioline). Primers were designed using primer3 and sequences are available in Supplementary Data 1.

## Western blot analysis iCAFs and myCAFs

Primary iCAFs and myCAFs were sorted from WEPtn-derived ILCs and after sorting directly lysed into lysisbuffer (1% Triton-X100, 150 mM NaCl, 50 mM Tris pH7.6, 1% sodium deoxycholate, 2 mM sodium orthovanadate and 1x complete protease inhibitor (Roche)). Samples were subjected to SDS-PAGE and western blotting on 0.2 um nitrocellulose membranes (Biorad, #1620112). Membranes were blocked in 5% non-fat milk in TBST and incubated overnight with the following antibodies: complement C3 (Abcam, ab200999) 1:1000. CD26 (Abcam, ab187048) 1:1000, TNC (Abcam, ab108930) 1:1000, TGFb1 (Abcam, ab179695) 1:1000, SMA (Sigma-Aldrich, clone 1A4) 1:1000 and Actin (Sigma-Aldrich, A5441) 1:2000. All primary antibodies were diluted in 5% non-fat milk in TBST. Secondary antibodies used for detection were diluted 1:2000 in 5% milk in TBST (goat-anti rabbit-HRP (Dako, P0448) and rabbit-anti mouse-HRP (Dako, P0260)).

## Transwell assay

All fibroblast recruiting transwell assays were performed in 24-well set-up using Corning inserts (8.0 um pores, transparent PET membrane, 1 × 10⁵ pores per cm²) and companion plates. Inserts were coated with growth-factor reduced matrigel (Corning) diluted in serum-free medium (DMEM) to a concentration of 0.25 mg/ml protein. Tumor cells or CM derived from tumor cells (1 × 10⁶ cells per 10 cm dish in serum-free medium for 24 h) were placed in the bottom wells. In each well 1 × 10⁵ tumor cells were plated. Twenty-four hours after plating the culture medium was replaced by serum-free medium (DMEM/F12). Coated inserts were placed in bottom wells and 5 × 10⁴ fibroblasts were plated in the inserts in DMEM + 0.2% FCS. After 24 h the inserts were harvested, cleared of cells remaining in the top compartment using cotton swaps and fixed in ice-cold methanol for 5 min. Next the inserts were stained with crystal violet (0.5% in 2:1 H2O:methanol) for 5 min. Inserts were cleaned in tap water and left to dry prior to imaging.

## Collagen contraction assay

Cells were plated in 1 ml gel composed of collagen (5 mg/ml):BME:medium (2:1:1) in a 24-well suspension plate and left to solidify at 37 °C, 5% CO₂ for 1 h. After incbuation, the gels were lifted from the plate and transferred to stainless-steel grids in a 6-well plate, allowing the gels to contract over a period of 3 days. Next the gels were fixed in 4% formalin and images were taken. The experiments with EpCAM−, EpCAM+, CD49f+ tumor cells and tdTomato+ CAFs were performed with 4 × 10⁵ cells per gel. The experiments with primary iCAFs and myCAFs (CD34+ vs. CD34− CAFs) were performed with 4 × 10⁵ cells per gel. The collagen contractions with primary fibroblasts and tumor cells were performed with either 4 × 10⁵ fibroblasts alone, 1 × 10⁵ tumor cells alone or 4 × 10⁵ fibroblasts co-plated with 1 × 10⁵ tumor cells.

## Organotypic invasion assay

CD26− or CD26+ NFs were plated in 1 ml gel composed of collagen (5 mg/ml):BME:medium (2:1:1) in 24-well suspension plates. In total, 4 × 10⁵ fibroblasts were plated in each gel and left to solidify at 37 °C, 5% CO₂ for 1 h. Next tumor cells were plated on fibroblast-containing or empty gels at a density of 1 × 10⁵ cells. One day after plating, the tumor

cells were covered with 100 ul BME:medium (1:1) to prevent disruption of the tumor cells monolayer during transfer. One hour after sealing the tumor cells, the gels were lifted from the plate and transferred to stainless-steel grids in a 6-well plate, allowing the gels to be surrounded by medium. The gels were cultured for 1 week in a 1:1 mix of tumor cell culture medium and fibroblast culture medium (hereafter referred to as mixed medium) in the absence or presence of Marimastat (100 nm) or neutralizing antibodies against CXCL12 (100 ug/ml). Medium was changed after 3 days. One week after plating the gels were harvested and fixed in 4% formalin and processed to FFPE slides for HE and EpCAM staining.

## Splenocyte recruitment assay

CM was derived from mono- or co-cultures of CD26− NFs, CD26+ NFs or tumor cells. CD26− or CD26+ NFs were plated in a 6-well plate at a density of $2 \times 10^5$ cells. Tumor cells (WEPtn derived) were plated in a 6-well plate at a density of $5 \times 10^4$ cells. For the co-cultures $2 \times 10^5$ fibroblasts were plated together with $5 \times 10^4$ tumor cells. All conditions were cultured in mixed medium for 3 days. Spleens were harvested from non-tumor bearing mice between 10 and 16 weeks of age and chopped into small pieces and digested briefly (15 min) in digestion mix, filtered and cleared of red blood cells (see section "flow cytometry analysis"). The splenocytes were resuspended in mixed medium and counted. All splenocyte recruitment assays were performed in 6-well set-up using Corning inserts (8.0 um pores, transparent PET membrane, $1 \times 10^5$ pores per cm²) and companion plates. Inserts were coated with growth-factor reduced matrigel (Corning) diluted in mixed medium to a concentration of 0.5 mg/ml protein. Coated inserts were placed in bottom wells with CM and $5 \times 10^5$ splenocytes were plated in the inserts in the presence or absence of neutralizing antibodies (also in bottom compartment): anti-CXCL12 (R&D systems, clone 79014, 100 ug/ml) or anti-CXCL2 (Thermo Fisher, clone 40605, 50 ug/ml). After 24 h the inserts were carefully removed and the bottom well contents were collected and spun down (300 g, 5 min). Pellets were resuspended in FACS buffer and stained using the following antibodies: CD11b-APC, CD3-BUV395 and B220-PE-Cy7. All samples were analyzed on the LSRII from BD biosciences. FlowJo v10 software was used for analysis.

## Cytokine array

WEPtn-derived tumor cells (13-MCB-17) were co-cultured with either CD26− NFs or CD26+ NFs at a density of $2.5 \times 10^5$ tumor cells with $1 \times 10^6$ fibroblasts in a 10 cm dish in mixed medium for 3 days. CM was harvested and spun at $2000 \times g$ for 5 min at 4 °C to pellet any cells or debris. The supernatant was used for the cytokine array (Proteome Profiler Mouse Cytokine Array Kit, Panel A, R&D systems) according to the manufacturer's protocol.

## Gene expression analysis

Based on the number of cells homogenized in the RLT buffer (79216, Qiagen), the total RNA was isolated using the RNeasy Mini Kit (74106, Qiagen) and RNeasy Micro Kit (74004, Qiagen), including an on-column DNase digestion (79254, Qiagen), according to the manufacturer's instructions. Quality and quantity of the total RNA was assessed on the 2100 Bioanalyzer instrument following manufacturer's instructions "Agilent RNA 6000 Nano" (G2938-90034, Agilent Technologies) and "Agilent RNA 6000 Pico" (G2938-90046, Agilent Technologies). Total RNA samples having RIN > 7 were subjected to TruSeq stranded mRNA library preparation, according to the manufacturer's instructions (Document # 1000000040498 v00, Illumina). The stranded mRNA libraries were analyzed on a 2100 Bioanalyzer instrument following the manufacturer's protocol "Agilent DNA 7500 kit" (G2938-90024, Agilent Technologies), diluted to 10 nM and pooled equimolar into multiplex sequencing pools for sequencing on the HiSeq 2500 and NovaSeq 6000 Illumina sequencing platforms. HiSeq 2500 single-end sequencing was performed using 65 cycles for Read 1, 10 cycles for Read i7, using HiSeq

SR Cluster Kit v4 cBot (GD-401-4001, Illumina) and HiSeq SBS Kit V4 50 cycle kit (FC-401-4002, Illumina). NovaSeq 6000 paired-end sequencing was performed using 54 cycles for Read 1, 19 cycles for Read i7, 10 cycles for Read i5 and 54 cycles for Read 2, using the NovaSeq6000 SP Reagent Kit v1.5 (100 cycles) (20028401, Illumina). Differential gene expression was determined using the R-package DESeq2[66] with the following criteria: FDR-corrected $p$ value <0.05 and log2FoldChange > 2 (up-regulated) or log2FoldChange < −2 (down-regulated). Hierarchical clustering was performed using Euclidean distance. For the pathway analyses a Fisher's exact test was used with the KEGG or GO geneset from MSigDB. Single sample gene set enrichment was calculated with the R-package Singscore[67].

## Single-cell transcriptomics of fibroblasts from WEPtn mice

$Cdh1^{F/F};Pten^{F/F}$ mice (EPtn, 7–8 weeks of age) were intraductally injected with lentivirus expressing Cre-recombinase to induce tumor formation or PBS as controls. Cre-injected glands were harvested 6, 12 or 18 weeks after injected. PBS-injected glands were harvested 12 weeks after injection. Glands/tumors were processed into a single cell suspension as described in the section "flow cytometry analysis". Samples were stained using the following antibodies: CD31-APC, CD45-AF700, EpCAM-FITC and CD49f-PE-Cy7. Fibroblasts were considered negative for all mentioned markers. Sorted fibroblasts were spun down and frozen in FCS + 10% DMSO until use. On day of scRNA sequencing, samples were thawed, checked for viability by flow cytometry (percentage of dapi-negative cells was above 75% for all samples). Cells were resuspended 1000 cells/ul in 1xPBS containing 0.04% weight/volume BSA (400 ug/ml) and for each sample the Chromium Controller platform of 10X Genomics was used for single cell partitioning and barcoding. Per sample each cell's transcriptome was barcoded during reverse transcription, pooled cDNA was amplified and Single Cell 3′ Gene Expression libraries were prepared according to the manufacturer's protocol "Chromium Next GEM Single Cell 3′ Reagent Kits v3.1" (CG000204, 10X Genomics). All four Single Cell 3′ Gene Expression libraries were quantified on a 2100 Bioanalyzer Instrument following the manufacturer's protocol "Agilent DNA 7500 kit" (G2938-90024, Agilent Technologies). These Single Cell 3′ Gene Expression libraries were combined to create one sequence library pool which was quantified by qPCR, according to manufacturer's protocol "KAPA Library Quantification Kit Illumina® Platforms" (KR0405, KAPA Biosystems). The NextSeq 550 Illumina sequencing system was used for paired end sequencing of the Single Cell 3′ Gene Expression libraries at a sequencing depth of ~18,000 reads pairs/cell. NextSeq 550 paired end sequencing was performed using 28 cycles for Read 1, 8 cycles for Read i7 and 56 cycles for Read 2, using NextSeq 500/550 High Output Kit v2.5 (75 Cycles) Reagent Kit (PN 20024906, Illumina). Cellranger version 4.0.0 in aggr mode was used with reference data mm10-2020 to calculate the counts for the 4 samples together. The resulting filtered_feature_bc_matrix.h5 file was loaded into scanpy version 1.7.1. Only cells that had minimum 200 genes were kept and only genes that were present in minimum 3 cells were kept. To filter out doublets and damaged cells only cells with genes by counts between 1000 and 4000 and with a mitochondrial percentage of less than 20% were kept. These cutoffs were based on distribution plots. The data was normalized and log-transformed. A neighborhood graph of observations was computed with 40 PC's (retaining 85% of the variance) and 20 local neighbors with method UMAP. Followed by clustering with the leiden algorithm with resolution 0.5 and visualized in a UMAP. Remaining clusters that contained expression of the marker genes (Cd31, Cd45, Epcam, Cd49f) used for sorting were excluded from the data. A neighborhood graph of observations was again computed, this time with 30 PC's (retaining 85% of the variance) and 20 local neighbors with method UMAP. Followed by clustering with the leiden algorithm with resolution 0.3 and visualized in a UMAP. The resolution was chosen based on having the highest silhouette score resulting in 4 clusters. Differential expression 1 vs. the

rest was done on the clusters with the Wilcoxon method and for the trajectory analysis PAGA was used[68].

## Single-cell transcriptomics of tumors from WB1P and WB1P-Myc mice

End-stage WB1P and WB1P-Myc tumors were digested into a single cell suspension as described in the section "Flow cytometry analysis". Single cell suspensions of the tumors were sorted by FACS to isolate live single cells (FSC-A x FSC-H to define singlets, dapi-negative cells to define live cells). After sorting the cells were subjected to Dropseq RNA sequencing according to the protocol of Macosko et al.[69] using WB1P cells (275 cells/ul) and WB1P-Myc cells (270 cells/ul) both in 1 x PBS + 0.01%BSA. Dropseq beads "MACOSKO-2011-10" were purchased from ChemGenes. Ready-made Drop-Seq microfluidic devices were purchased from Nanoshift. During droplet generation, of both samples multiple 5 min fractions of droplets were collected and based on droplet quality assessment for each sample 4 fractions were selected for further processing according to "Drop-seq laboratory protocol v3.1". After breakage, reverse transcription and Exocuclease I treatment, of each fraction the cDNA was amplified in triplicate PCR reactions. After PCR the amplified cDNA products were pooled, cleaned by a 0.6X AMPure XP bead cleanup and quantified on a 2100 Bioanalyzer Instrument following the manufacturer's protocol "Agilent High Sensitivity DNA Kit" (G2938-90321, Agilent Technologies). This procedure was done twice for every fraction selected. The amplified cDNA product was concentrated by speedvac and all was used as input for the final Nextera XT library preparations. All Libraries were quantified on a 2100 Bioanalyzer Instrument following the manufacturer's protocol "Agilent High Sensitivity DNA Kit" and diluted to 10 nM before paired-end sequencing on the MiSeq and HiSeq2500 Illumina sequencing platforms. Sequencing was performed using 25 cycles for Read 1, 8 cycles for Read i7 and 117 cycles for Read 2, using MiSeq Reagent Kit v3 (150-cycle) (MS-102-3001, Illumina), HiSeq PE Rapid Cluster Kit v2 (PE-402-4002, Ilumina) and HiSeq Rapid SBS Kit v2 (200 cycles) (FC-402-4021, Illumina). Sequencing was performed on the MiSeq and HiSeq2500 Illumina sequencing platforms. Counts from the dropseq data were created with dropseq-tools 1.12. The resulting files were loaded into scanpy version 1.7.1. Only cells that had minimum 200 genes were kept and only genes that were present in minimum 3 cells were kept. To filter out doublets and damaged cells only cells with genes by counts between 450 and 4000 and with a mitochondrial percentage of less than 20% were kept. These cutoffs were based on distribution plots. The data was normalized and log-transformed. A neighborhood graph of observations was computed with 30 PC's (retaining 85% of the variance) and 20 local neighbors with method UMAP. Followed by clustering with the leiden algorithm with resolution 0.3 and visualized in a UMAP. The fibroblasts were selected by selecting the cluster expressing Col1a1 and Col1a2. A neighborhood graph of observations was computed with 40 PC's (retaining 85% of the variance) and 20 local neighbors with method UMAP. Followed by clustering with the leiden algorithm with resolution 0.3 and visualized in a UMAP.

## Single-cell transcriptomics of tumors from WEPtn and WEH1047R mice

ILC tumors were isolated from 20-week old WEPtn and WEH1047R mice ($n = 3$ mice per model) and digested into a single cell suspension as described in the section "Flow cytometry analysis". Single-cell suspensions were labeled with barcoding hashtags from biolegend (TotalSeq anti-mouse hashtag 1, 2 and 3, directed at CD45 and MHC-I clone M1/42 and 30-F11) to allow for separation of the three replicates used for each model. Next the cells were sorted to isolate live single cells (FSC-A x FSC-H to define singlets, dapi-negative cells to define live cells). After sorting the cells were resuspended 1000 cells/ul in 1xPBS containing 0.04% weight/volume BSA (400 ug/ml) and for WEPtn and WEH1047R the hashtagged replicates were pooled and processed as

described in the section "Single-cell transcriptomics on fibroblasts from WEPtn mice".

## Data analysis

The following programs and versions were used for the analysis of single-cell and bulk transcriptomics: Matplotlib 3.3.4. Numpy 1.19.2. Pandas 1.2.3. Scanpy 1.7.1. Sinfo 0.3.1. IPython 7.21.0. Jupyter_client 6.1.7. Jupyter_core 4.7.1. Notebook 6.2.0. Python 3.7.10 (default, Feb 26 2021, 10:16:00) [Clang 10.0.0]. Darwin-19.6.0-x86_64-i386-64bit 8 logical CPU cores, i386. Session information updated at 2021-03-09 14:06. FlowJo v10.8.0. Qupath 0.3.0. ImageJ 1.53t. Leica Application Suite X (LASX) 3.7.6.

## Statistical analysis

The statistical analyses used throughout the manuscript have been indicated in the figure legends.

## Reporting summary

Further information on research design is available in the Nature Portfolio Reporting Summary linked to this article.

## Data availability

The raw sequence data generated in this study are available at the Gene Expression Omnibus database under the following accession numbers: GSE205263 and GSE214933. The publicly available human single-cell transcriptomics dataset[36] used in this study is available in the Gene Expression Omnibus database under accession code GSE161529. The publicly available laser-microdissected human ILC and IDC data[35] used in this study are available in the Gene Expression Omnibus database under accession codes GSE148398 (ILC) and GSE68744 (IDC). The publicly available single-cell transcriptomics dataset of murine PDAC[5] used in this study is available in the Gene Expression Omnibus database under accession code GSE129455 (https://www.ncbi.nlm.nih.gov/geo/query/acc.cgi). The remaining data are available in Supplementary Data 1 and in the source data file. Source data are provided with this paper.

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

## Acknowledgements

This research was funded by Oncode Institute (J.J.), the Dutch Research Council (NWO, Veni grant 016.196.120, J.M.H.) and the Dutch Cancer Society (KWF, grants NKI 2015-7589 (M.C.B.) and NKI 2021-13751, J.M.H.). We would like to thank the people at the Genomics Core Facility of the Netherlands Cancer Institute, especially Wim Brugman, for all the RNA isolations, RNA library preps and sequence runs presented in this study.

## Author contributions

J.M.H. performed and designed the experiments and wrote the paper together with J.J. R.d.B. performed all bioinformatic analyses on the single-cell transcriptomics datasets and other in vitro and in vivo RNA sequencing experiments. E.v.d.B. and A.P.D. assisted with the animal experiments and performed tumor volumetric measurements on all mice. E.W. was responsible for genotyping of the mice used within these experiments and assisted with in vitro experiments. T.F. assisted with the analysis of tumors from the transplantations in the *En1-Cre;mTmG* mice. E.B. performed the iCAF and myCAF signature analyses on the human ILC and IDC laser microdissected tumor specimens. C.S.B. prepared the samples for single-cell transcriptomics of the WB1P and WB1P-Myc tumors. E.M.P. assisted with immunofluorescent stainings of CAF subtypes in the different mouse models. M.N. and I.d.R. performed the single-cell transcriptomics and quality control of RNAseq data. F.v.D. assisted with FACS sorting of tumor samples and co-cultures for sequencing. S.K. generated the WEPtn mouse model. R.K. was head of sequence facility and assisted in the design of single-cell transcriptomics experiments. V.G.B. published and provided the human ILC and IDC datasets. C.L.G.J.S. performed whole mount analyses on *En1-Cre;mTmG* transplanted mice and wildtype mice. M.C.B. initiated the CAF research in WEPtn mice. J.J. is head of the Molecular Pathology department and wrote the paper together with J.M.H.

## Competing interests

The authors declare no competing interests.
