## [Peer Review File · Nature Communications]

CD26-negative and CD26-positive tissue-resident fibroblasts contribute to functionally distinct CAF subpopulations in breast cancerReviewers' Comments:

Reviewer #1:

Remarks to the Author:

The ms entitled "CD26-negative and CD26-positive tissue-resident fibroblasts contribute to functionally distinct CAF subpopulations in breast cancer" by Houthuijzen et al. addresses the cellular origin of CAFs in mouse models of luminal and triple-negative mammary cancer, and focuses further on CD26- and CD26+ stromal cells. It is concluded that mammary normal fibroblasts may generate CD26- and CD26+ CAF subpopulations, of which the CD26+ cells exhibit protumorigenic properties. The experimental approach is contemporary and basically sound and represents a considerable effort, but this reviewer has several major and minor points of concern that needs to be addressed:

General:

- 1) The use of "breast cancer" in the title and elsewhere suggests that the findings in the mouse is applicable to human breast and human breast cancer. For instance, last sentence in the abstract: "Together, our data show that CD26+ and CD26- NFs transform into distinct subpopulations in breast cancer". While the findings may be of relevance for human breast cancer, such mixing of mouse mammary and human breast gland is misleading. More so, since terminal duct lobular units, which in the human normal breast harbors CD26- fibroblasts (ref. 16), are not present in the mouse gland. I suggest that "mammary" is used instead of "breast" in the title, abstract, last sentence in Introduction etc., and that the text is rephrased to suggest – rather than show – that the different CAF subpopulations are relevant in breast cancer.
- 2) Some papers are not correctly cited. For instance, at p. 3 it is stated that "through a series of in vivo transplantations LeBleu...(ref. 13)". This is not correct. The method used is unilateral ureteral obstruction. P. 4. The original reference on the mouse models is Annunziato, Nat. Commun., 2019, not ref. 24. P. 6 "...BM-MSCs contribute to the pool of breast cancer (26, 10, 14, 12.)" Ref. 12 shows that endothelial cells contribute and does not include BM-MSCs. The authors are advised to check all references.
- 3) If in line with the policy of the journal, it would be helpful to indicate in the text when reviews rather than original papers are cited, for instance p. 2., ref. 2.
- 4) Say "in culture" instead of "in vitro" throughout. "In vitro" also covers experiments without cells. Could be protein-protein interactions etc.
- 5) Abbreviate fibroblasts as "fib" instead of "fibro 's".

Introduction:

- 6) Please refrain from using slang: "bad guys". Also, say "previous" or "several" instead of "various" studies, here and elsewhere in the ms.
- 7) When referring to previous studies that have identified CAFs with tumor-restraining properties, the study by Hutton et al., Cancer Cell 2021, should be included. It would also be appropriate to include this study in the Discussion, since the findings by Hutton et al. support the argument by others for phenotypic states rather than defined lineages. Also, they find that CD26 is equally expressed in myCAFs and iCAFs.
- 8) The sentence "The arrival of single cell genomics has uncovered..." implies that heterogeneity among CAFs and diverse cellular origins have not been addressed previously. Several studies using other methods have addressed myofibroblast origin (from resident fibroblasts including perivascular fibroblasts, smooth muscle cells, pericytes etc.) in breast and other tissues. The authors are advised to search the literature and look up earlier studies. See for example, Ronnov-Jessen et al., J. Clin. Invest. 1995 and Sugimoto et al., Cancer Biol. Ther. 2006.

Results:

- 9) P. 4: "...we used our clinically relevant GEMMs..". Please explain how a mouse model can be clinically relevant.
- 10) ILC and TNBC lesions were compared regarding stromal reaction. Since CD26 is pursued as a marker for CAF subtype, the protein expression pattern of CD26 among CAFs should be addressed. Also, since a functional significance of CD26 is sought for, any information at the protein level rather than the RNA level would be of interest. Likewise, a more detailed description of the distribution of

CD26+ and CD26- fibroblasts in the normal gland should be given. In the human breast, the two fibroblast subtypes localize to ducts and lobules, respectively. According to 4C, between 45 and 72% of the EpCAM-/CD45 (-? - move separation)/CD31-/CD49f- cells are CD26-, and a staining of periductal CD26+ fibroblasts is included in Fig. 4B, but where do the CD26- come from? This is of crucial importance for the interpretation of the results regarding comparison of CD26+ and CD26-normal-derived fibroblasts (NFs).

11) In fact, a major point is the definition of the fibroblasts. In Figs. 1 and 2, fibroblasts are defined as PDGFRb+, but later fibroblasts (Figs. 3 and 4) are gated as EpCAM-/CD45-/CD31-/CD49f- only. Why this change in approach?

Also, tdTomato-negative cells that lacked epithelial cells believed to be fibroblasts did not express PDGFRb, p. 6 (Suppl. Fig. 3A). Others have found that PDGFRb decreases with tumor development and is not uniformly expressed among CAFs (Venning et al., J. Exp. Clin. Cancer Res.). The paper should be cited and discussed.

12) The statement "Since these findings contradict ...(ref. 14)" needs to be clarified. In which way are the findings contradictory? Ref. 14 finds that BM-CAFs do not express PDGFRa as opposed to resident CAFs and thus, can distinguish the two origins.

13) Suppl. Fig. 5 D and E and p. 8: "Again, BM-MSCs were not the main contributors to the population of CAFs, as the majority of the CAFs in the resulting mammary tumors were tdTomato-positive". Fig. 4D shows the tdTomato expression among each of the CD45+, CD31+ and PDGFRb+ populations. Is it the higher frequency of tdTomato+ among PDGFRb+ cells that leads to this conclusion? If the transplanted MSCs lacked fluorescence, where do the tdTomato+ /CD45+ cells (approximately 20%) come from?. Please clarify and rewrite.

Also, an analysis similar to the one done for the normal gland in Suppl. Fig. 6 B would be helpful.

14) The presentation of results in Fig. 3 is of major concern. Cluster 2 is said to represent iCAFs and cluster 4 myCAFs, but who is who in Fig. 3F and 3G? The iCAF signature now corresponds to cluster 4 and the other way around. Until this is fixed, it is not possible to assess whether the following results presented are correct ! It is claimed that there is a mix of CAFs with iCAF and myCAF signatures in the tumors, but is this really true then?

15) The authors subsequently focuses on CD26, and say that "The presence of two distinct fibroblast populations that could be distinguished from each other by CD26 expression has previously been reported in normal human and mouse mammary tissue (refs. 16, 31)", p. 12. Firstly, it is somewhat understated as the two previous studies found an immune-related signature in CD26+ fibroblasts, and that Dpp4+ Fib_0 represented "INFLAMMATORY RESPONSE", so to claim CD26+ to contribute iCAFs as "our" hypothesis is not entirely appropriate. Secondly, how do the cluster 3 CD26+ relate to these previously described iCAF (?) populations? In the present study – provided that Fig. 3C is correct of course...- cluster 2 rather than cluster 3 represents iCAFs. At the very least this should be reflected upon and discussed.

16) Fig. 4A and D "...confirming the absence of CD26 in myCAFs and low CD26 expression in iCAFs". So now, iCAFs express low CD26. What is low compared to - og +? And then there is a shift back to CD26- and CD26+. This is very confusing!

17) The stroma of ILC and TNBC comprises both iCAF and myCAF signatures.

P. 15: "Together, these findings indicate that CD26+ are at the origin of pro-tumorigenic CAFs."

Again, others have found that CD26low normal breast fibroblasts have more genes in common with tumor stroma than CD26high fibroblasts (ref. 31), and it has been argued that in pancreatic ductal adenocarcinoma, CD26 expression indicates phenotypic states (Hutton et al. Cancer Cell, 2021). There is a need for further distinction, reflection and discussion in relation to mouse versus human. In the human gland CD26+ fibroblasts surround the ducts and CD26- surround the terminal duct lobular units (which the mouse does not have), and duct- and TDLU-derived breast carcinomas are known to exhibit unique histological appearances and clinical outcomes (Tabar et al. Breast Cancer, 2014).

Reviewer #2:

Remarks to the Author:

The study by Houthuijzen et al was designed to uncover the origin of cancer-associated fibroblasts in mammary tumors using advanced GEMM models of breast cancer. Based on various transplantation experiments, the authors conclude that CAFs originate from resident mammary gland fibroblasts. Through single cell transcriptomics, subsets of normal fibroblasts with or without expression of CD26 are identified to give rise to iCAF and myCAF, respectively. Taken together, the study sheds light on the complexity of breast CAFs and may suggest a path towards precision targeting. While the study is well executed technically, a number of issues remain to be resolved before publication is warranted.

1. The authors are commended on the design of the illuminating transplantation experiments. However, the assessment of CAF abundance by FACS should be complemented by visualization in tissue. In particular, the conclusion by the authors that epithelial cells that have gone through an EMT do not constitute a significant part of the CAF pool may be flawed, since the negative selection marker CD49f has been proven to be expressed by EMT-derived CAFs (see e.g. <https://doi.org/10.1002/stem.791> and <https://doi.org/10.1038/s41467-018-07582-3>). To support this conclusion, the authors should provide alternative evidence.
2. Cluster 4 CAFs are designated as myCAFs by the authors based on GO analysis of a 25-gene signature of differentially expressed genes. However, the GO categories identified are mostly related to ECM, not contraction. The authors should demonstrate the superior contractile ability of Cluster 4 myCAFs over Cluster 2 iCAFs. Also, differential gene expression analysis between all four clusters (presumably two NF clusters and 2 CAF clusters) may hide functional relationships between the clusters since the algorithm is looking for unique properties of each cluster. The authors should explore other clustering resolutions and visualize this with a clustree in order to further inform the differential gene expression analysis.
3. In Fig. 3C, the label for the Y-axis is absent. What do the different sets of time points represent? A matrix with the expression of each set of top 10 DEGs in all CAF clusters would have been helpful. Also, the iCAF/myCAF cluster signatures are not very convincing when applied to other datasets. How would they look visualized on the original feature plot? And on other single cell datasets for breast cancer?
4. In the cell culture experiments using isolated CD26- and CD26+ fibroblasts, the authors need to demonstrate that these truly represent stable lineages by assessing CD26 expression over time. Also, while gene expression signatures appear to be regulated in co-culture experiments, these say nothing about functionality. Are CD26- fibroblasts conditioned by ILC cells superior in contractile abilities? Similarly, do CD26+ fibroblasts modulate immune cell function following tumor cell conditioning?
5. The fact that the authors need to use alternative culturing conditions to induce CD26- and CD26+ fibroblasts into myCAFs and iCAFs, respectively, detracts from the credibility of the proposed model. In vivo co-transplantation experiments of differentially labeled CD26- and CD26+ fibroblasts and subsequent analysis of the fate of these cells as they convert into CAFs would be more illuminating and necessary to support the model.

Reviewer #3:

Remarks to the Author:

In their manuscript, Houthuijzen and colleagues identify normal tissue fibroblasts as the main source of CAFs in both invasive lobular breast cancer and triple negative breast cancer models. They further characterize fibroblast heterogeneity in normal tissue and in the breast cancer TME and identify CD26 as a marker of two subsets of fibroblasts that are predicted to give rise to distinct populations of CAFs. Based on in vitro modelling and in vivo experiments the authors suggest that CD26+ fibroblasts primarily give rise to inflammatory CAFs, while CD26- fibroblasts develop into myofibroblastic CAFs. Overall, the manuscript is well written and logically structured. The in vivo experiments which establish normal mammary tissue-resident fibroblasts as the main source of CAFs in breast cancer are novel and of interest to the field. The key challenges I see with the current version of the manuscript are:

Major

1) The En1-Cre;mTmG lineage tracing system which is intended to in vivo trace CD26+ and CD26- fibroblasts is not suited for this purpose due to the non-specific expression of En1. Therefore, the authors rely on CD26 protein measurements to infer CAF origin in vivo and not the reporter. This is a) very time intensive to follow for the reader without any benefit of the reporter system and b) insufficient evidence for the conclusion that "in vivo tracing and in vitro studies revealed the transition of CD26+ and CD26- NF populations into inflammatory CAFs (iCAFs) and myofibroblastic CAFs (myCAFs), respectively". Additionally, in the in vitro systems the observed CAF subtype was highly dependent on the culture system used and therefore these data do also not allow a direct conclusion on the proposed lineage relationships.

2) The authors attempt to untangle CAF origin and programs specifically induced in these cells in a tumor setting. Yet, their signatures for iCAFs and myCAFs remain convoluted and a mixture of origin and induced programs. This convoluted signature [derived from end-stage WB1P and WB1P-Myc tumors] biases the analyses for example in figure 6B: Half of the genes shown visually look already enriched in CD26+ vs CD26- normal fibroblast. As such, the conclusion "As expected, the iCAF gene signature was enriched in the CD26+ NFs cultured in tumor CM" is (visually) true, but partially because of the convoluted baseline expression program and not because TCM induced the expression of these genes in one or the other population (In fact, some of the iCAF genes seem lost with TCM in CD26+ fibroblasts). I would suggest the authors improve their CAF signatures using the single-cell data from their time course in ILC-tumor bearing mice to be able to make more reliable statements on genes induced by tumor conditioned media in CD26+/- cells respectively.

3) The computational analysis requires more robust statistics and partially lacks replication

- Some experiments seem to have no replicates (n=1), which does not allow to make reliable statements on significance of gene expression changes (e.g Fig 6 E, F not replicated for NF)
- In multiple paragraphs the authors make statements about enrichment even though no tests were performed. Often upregulation is inferred "visually". For example, why are no differential expression tests performed that support the statements on the results in figures 6B,C, E,F, K, L? Similarly: Why is there no statistical test performed for the values in 3H comparing scores in epithelium and stroma?
- The rows of the heatmaps are clustered, but no cluster dendrograms are provided in these heatmaps. For some figures the authors might even want to avoid clustering in order to allow comparisons between sub panels more easily (For example it's hard to compare between 6B and E due to different ordering of the genes).
- For some differential expression comparisons that were performed it is unclear what conditions were compared ("Hierarchical clustering of the top-100 differentially expressed genes" Top differentially expressed between which conditions?)
- The legends in multiple panels do not have a label for the color range. What is plotted (E.g. 4A, E, 6B,C, E, F, K, L,)? Z-scores? Of what? Log CPM? CPM?
- Essential information on the single-cell data is missing: Supplemental tables for markers of identified clusters, results of differential expression tests, gene set enrichment analysis, ..
- The computational single cell methods provided do not allow to fully understand the analyses performed. The only two sentences on downstream analysis after QC/filtering are given as: "Clustering was made using the Leiden algorithm. Trajectory analysis was performed using the PAGA[60]." For none of the analyses performed any parameters are provided, what dimensionality reduction techniques were used, if the analysis was focused on a set of most variable genes, ... All these things would be quite essential to understand how this data has been processed.

Minor

Fig 3 F,G: iCAF/myCAF labels flipped; I do not understand what logFC is shown here. Are these z-cores or LogFCs? If logFC compared to what?

3I-J: What the authors annotate as CAFs are Rgs5+Acta2+ cells and likely pericytes

SF12 These heatmaps are really hard to interpret. In it looks like CD26 is higher in CD26- fibroblasts with TCM than in CD26+ NT fibroblasts (which seem to have low CD26 expression and CD26 expression increases with conditioned media). Is this data transformed in some kind of way or am I

interpreting the figure incorrectly? Seems counterintuitive, especially also when looking at Dpp4 expression comparing CD26+ and CD26- NT fibroblasts. Seems like they express Dpp4 at the same (low) level?

The authors do not attempt to infer relationships between CD26+ and - cells in normal tissue. Are CD26- fibroblasts derived from a CD26+ cell or vice versa? Can the authors use velocity analyses to make computational predictions?

Rebuttal Nature Communications manuscript NCOMMS-22-19756-T

Textual changes are highlighted in yellow within the manuscript text and additional experimental data and its results are highlighted in blue.

Reviewer #1, expertise in the origin of CAFs and breast cancer TME:

The ms entitled "CD26-negative and CD26-positive tissue-resident fibroblasts contribute to functionally distinct CAF subpopulations in breast cancer" by Houthuijzen *et al.* addresses the cellular origin of CAFs in mouse models of luminal and triple-negative mammary cancer, and focuses further on CD26- and CD26+ stromal cells. It is concluded that mammary normal fibroblasts may generate CD26- and CD26+ CAF subpopulations, of which the CD26+ cells exhibit protumorigenic properties.

The experimental approach is contemporary and basically sound and represents a considerable effort, but this reviewer has several major and minor points of concern that needs to be addressed:

General:

1) The use of "breast cancer" in the title and elsewhere suggests that the findings in the mouse is applicable to human breast and human breast cancer. For instance, last sentence in the abstract: "Together, our data show that CD26+ and CD26- NFs transform into distinct subpopulations in breast cancer". While the findings may be of relevance for human breast cancer, such mixing of mouse mammary and human breast gland is misleading. More so, since terminal duct lobular units, which in the human normal breast harbors CD26- fibroblasts (ref. 16), are not present in the mouse gland. I suggest that "mammary" is used instead of "breast" in the title, abstract, last sentence in Introduction etc., and that the text is rephrased to suggest – rather than show – that the different CAF subpopulations are relevant in breast cancer.

We thank the reviewer for this suggestion. We have clearly stated in the revised manuscript which experimental data is derived from our breast cancer mouse models and which data is derived from human patient material. We agree with the suggested change regarding the last sentence of the abstract and therefore changed this to: 'Together, our data suggest that CD26+ and CD26- NFs transform into distinct subpopulations in mouse models of breast cancer'. Additionally, we have changed the title to: 'CD26-negative and CD26-positive tissue-resident fibroblasts contribute to functionally distinct CAF subpopulations in mouse models of breast cancer'.

2) Some papers are not correctly cited. For instance, at p. 3 it is stated that "through a series of *in vivo* transplantations LeBleu...(ref. 13)". This is not correct. The method used is unilateral ureteral obstruction. P. 4. The original reference on the mouse models is Annunziato, Nat. Commun., 2019, not ref. 24. P. 6 "...BM-MSCs contribute to the pool of breast cancer (26, 10, 14, 12.)" Ref. 12 shows that endothelial cells contribute and does not include BM-MSCs. The authors are advised to check all references.

We understand the potential confusion regarding reference 13 (reference 16 in revised manuscript). By stating 'through a series of *in vivo* transplantations' we were pointing towards the bone marrow transplantations performed in the paper, rather than any

transplantations done to induce the disease model studied. This sentence has been rephrased to: 'Through bone marrow transplantations and fibroblasts lineage tracing models LeBleu and colleagues showed that fibroblasts present in fibrotic kidney disease consisted of locally activated fibroblasts, BM-MSCs and trans-differentiated endothelium-derived fibroblasts'. This and other textual changes in the revised manuscript are highlighted in yellow.

The reviewer is partly correct regarding the reference for the WB1P and WB1P-Myc mouse models. The reference of Annunziato *et al.* has been added to refer to the WB1P and WB1P-Myc mouse models. We believe the reference to Zimmerli *et al.* (ref 27 in revised manuscript) is still relevant as this article highlights the differences observed in the tumor microenvironments of WB1P and WB1P-Myc tumors and we believe this may be relevant for readers.

The reference to Zeisberg *et al.* Cancer Res. (ref 12) has been removed from the sentence on page 6 and now reads: Previous reports have shown that bone marrow-derived mesenchymal stem cells (BM-MSCs) can contribute to the pool of CAFs in breast cancer[27,10,14]. In addition, all other references have been thoroughly checked.

3) If in line with the policy of the journal, it would be helpful to indicate in the text when reviews rather than original papers are cited, for instance p. 2., ref. 2.

Although this is not a requirement for the journal, we agree with the reviewer that this can be helpful to the readers. Therefore we indicated in the text which references are reviews (reference 1, 2, 11, 12, 20, 22 and 56 are reviews).

4) Say "in culture" instead of "*in vitro*" throughout. "*In vitro*" also covers experiments without cells. Could be protein-protein interactions etc.

We agree with the reviewer that '*in vitro*' may be too general when referring to experiments performed in cell culture. We changed '*in vitro*' to 'in culture' where this was appropriate and highlighted this in yellow in the manuscript text.

5) Abbreviate fibroblasts as "fib" instead of "fibro's".

We have changed the abbreviation 'fibro's', which was used in supplemental figures 1 and 2, to 'fibroblasts'.

Introduction:

6) Please refrain from using slang: "bad guys". Also, say "previous" or "several" instead of "various" studies, here and elsewhere in the ms.

We have rephrased the sentence: 'However, to label CAFs as the 'bad guys' within the TME is too simplistic' to: 'However, to consider CAFs solely as 'tumor-promoting' is too simplistic'. Also terminology like 'various' has been changed to 'previous' or 'multiple'. These changes are marked yellow in the text.

7) When referring to previous studies that have identified CAFs with tumor-restraining properties, the study by Hutton *et al.*, Cancer Cell 2021, should be included. It would also be appropriate to include this study in the Discussion, since the findings by Hutton *et al.* support the argument by others for phenotypic states rather than defined lineages. Also, they find that CD26 is equally expressed in myCAFs and iCAFs.

We thank the reviewer for this suggestion. We have added the reference of Hutton *et al.* and discussed the paper in the Discussion (see page 16). We believe that from this paper the conclusion cannot be drawn that CD26 is equally expressed in myCAFs and iCAFs. The authors find that CD105+ fibroblasts in pancreatic cancer support tumor growth whereas CD105- fibroblasts inhibit tumor growth. They state that CD26 expression (as well as other markers previously associated with iCAF and myCAFs) are equally expressed between CD105- and CD105+ fibroblasts. The authors are not comparing iCAFs versus myCAFs but CD105+ versus CD105- fibroblasts, and since the authors state that both CD105- and CD105+ fibroblasts can become iCAFs and myCAFs the comparison between CD105- and CD105+ is not the same as comparing iCAFs with myCAFs.

We performed single-cell transcriptomics on ILCs isolated from WEPtn and WEH1047R mice to further investigate the iCAF and myCAF gene signatures in the context of the entire tumor microenvironment rather than in fibroblasts only. Using these datasets, which were added to the manuscript in figure 4A-E and supplemental figure 8A-C, we were able to investigate *Eng* (gene encoding for CD105) expression. We found limited expression of *Eng* in the fibroblast clusters. However, the endothelial cell cluster displays more robust expression of *Eng*, consistent with literature describing CD105 as an activated endothelial cell marker (Cheifetz *et al.* J Biol Chem 1992 and figure 1A below). Next, we investigated CD105 protein expression by flow cytometry of normal mammary glands and WEPtn-derived tumors. We did not observe prominent CD105 protein expression in normal mammary fibroblasts or WEPtn-derived CAFs, but CD105 was expressed by a subset of CD31+ endothelial cells in normal mammary gland and enhanced CD105 expression was seen in endothelial cells within tumors (see Figure 1B-C below). The lack of robust CD105 expression in mammary fibroblasts means we cannot perform similar studies or comparisons as described by Hutton *et al.* Furthermore, this data emphasizes that, with regards to marker expression, pancreatic fibroblasts express different markers than mammary fibroblasts.

Figure I

Figure I: CD105 expression in normal mammary gland and ILC. A) Single-cell transcriptomics of WEPTn and WEH1047R derived ILCs (20 week old mice, n=3 per model). UMAP plots show relative expression of *Eng* (CD105). B) Flow cytometry analysis of normal mammary glands for CD105 expression. Mammary epithelial cells were defined as EpCAM+, immune cells as CD45+ and endothelial cells as EpCAM-/CD45-/CD31-. Fibroblasts were defined as EpCAM-/CD45-/CD31-. Two most right plots show expression of CD105 in endothelial cells and normal mammary fibroblasts. Within these plots the FMO control is shown in blue and the full stained sample is shown in red. C) Flow cytometry analysis of WEPTn-derived ILC for CD105 expression. Mammary epithelial cells were defined as EpCAM+, immune cells as CD45+ and endothelial cells as EpCAM-/CD45-/CD31-. Fibroblasts were defined as EpCAM-/CD45-/CD31-. Two most right plots show expression of CD105 in endothelial cells and cancer-associated fibroblasts. Within these plots the FMO control is shown in blue and the full stained sample is shown in red.

8) The sentence “The arrival of single cell genomics has uncovered...” implies that heterogeneity among CAFs and diverse cellular origins have not been addressed previously. Several studies using other methods have addressed myofibroblast origin (from resident fibroblasts including perivascular fibroblasts, smooth muscle cells, pericytes etc.) in breast and other tissues. The authors are advised to search the literature and look up earlier studies. See for example, Ronnov-Jessen *et al.*, J. Clin. Invest. 1995 and Sugimoto *et al.*, Cancer Biol. Ther. 2006.

We understand that this sentence may not do enough justice to the many papers that have indeed previously shown CAF heterogeneity independently of single cell transcriptomics. Therefore we included the following references: Sugimoto *et al.* Cancer Biol Ther. 2006 (ref 7), Bauer *et al.* Oncogene 2010 (ref 8) and Ronnov-Jessen *et al.*, J. Clin. Invest. 1995 (ref 6). In addition, we rephrased the sentence to: 'Multiple studies have uncovered CAFs as a heterogeneous cell population, an observation that has been further highlighted by the arrival of single cell transcriptomics, which have uncovered myofibroblastic or extracellular matrix (ECM) producing CAFs, antigen-presenting CAFs and inflammatory CAFs (5-8).' See page 2 (highlighted in yellow).

Results:

9) P. 4: "...we used our clinically relevant GEMMs..". Please explain how a mouse model can be clinically relevant.

The mouse mammary tumor models we have used in this study are based on genetic drivers found in human breast cancer and display a similar growth pattern, tumor microenvironment composition and histology as their human counterparts, making them relevant model systems for the study of human breast cancer development and progression. However, we agree that the term 'clinically relevant' may be confusing to readers and therefore we changed it to: 'established GEMMs that closely mimic human breast cancer histology' (see page 4, yellow highlighted).

10) ILC and TNBC lesions were compared regarding stromal reaction. Since CD26 is pursued as a marker for CAF subtype, the protein expression pattern of CD26 among CAFs should be addressed. Also, since a functional significance of CD26 is sought for, any information at the protein level rather than the RNA level would be of interest. Likewise, a more detailed description of the distribution of CD26+ and CD26- fibroblasts in the normal gland should be given. In the human breast, the two fibroblast subtypes localize to ducts and lobules, respectively. According to 4C, between 45 and 72% of the EpCAM-/CD45 (-? – move separation)/CD31-/CD49f- cells are CD26-, and a staining of periductal CD26+ fibroblasts is included in Fig. 4B, but where do the CD26- come from? This is of crucial importance for the interpretation of the results regarding comparison of CD26+ and CD26- normal-derived fibroblasts (NFs).

With regards to the expression of CD26 in NFs and CAFs, the manuscript contains flow cytometry data and whole mount analysis (figures 3I, 5B-D, 7B and supplemental figure 11) which all reflect protein expression of CD26 rather than mRNA. In our whole mount analysis we did not observe a clear localization of CD26+ NFs in adult mouse mammary glands, as CD26+ NFs were present both at terminal end buds and around ducts, in contrast to the tissue distribution described in human mammary tissue, where CD26- NFs reside around lobules and CD26+ NFs around ducts. These discrepancies between human and mouse are mentioned in the discussion (see page 15, yellow highlighted).

Efforts to understand the relationship between CD26- and CD26+ NFs in the adult mammary gland from our control mice in our single-cell transcriptomics dataset using velocity analysis suggests that CD26- NFs residing at the border between the clusters of

CD26⁻ and CD26⁺ NFs give rise to both populations (see figure II below). However, we believe that elucidation of the origin of CD26⁻ and CD26⁺ mammary NFs cannot be fully answered with our current data sets as this requires analysis during mammary gland development rather than in adult, steady-state settings or tumor settings and we feel this is beyond the scope of the manuscript. Nevertheless, data from *En1-Cre;mTmG* mice would suggest, since a portion of both CD26⁻ and CD26⁺ NFs are derived from Engrailed1-positive precursors, that at least part of these cells share a common neuroectoderm ancestor cell during embryonic development.

Figure II

Figure II: Velocity analysis of normal fibroblasts in the mammary gland. A) Velocity embedded stream plot of single-cell dataset of normal mammary fibroblasts (control time point only, non-tumor-bearing mice). B) Trajectory analysis using PAGA of normal mammary fibroblasts suggests that cluster 2, CD26⁻ fibroblasts on the border between CD26⁻ and CD26⁺ cluster, are at root of the two fibroblast subtypes. C) Pseudotime presentation of velocity analysis. 0 indicates early, 1 indicates late. D) Relative expression of *Dpp4*.

11) In fact, a major point is the definition of the fibroblasts. In Figs. 1 and 2, fibroblasts are defined as PDGFR β ⁺, but later fibroblasts (Figs. 3 and 4) are gated as EpCAM⁻/CD45⁻/CD31⁻/CD49f⁻ only. Why this change in approach? Also, tdTomato-negative cells that lacked epithelial cells believed to be fibroblasts did not express PDGFR β , p. 6 (Suppl. Fig. 3A). Others have found that PDGFR β decreases with tumor development and is not uniformly expressed among CAFs (Venning *et al.*, J. Exp. Clin. Cancer Res.). The paper should be cited and discussed.

We understand the potential confusion regarding the fibroblast definitions used throughout the manuscript. In line with others, we also find variable expression levels of PDGFR β in CAFs based on our single-cell transcriptomics (data added to supplemental figure 7B of revised manuscript). The main reason for using PDGFR β in the transplantation studies is that this marker is most widely expressed among both CAFs and normal fibroblasts in our models (compared to other markers like SMA, Fsp1 and podoplanin) and it was also the only marker we could use to reliably separate EMT tumor cells from fibroblasts (as shown in supplemental figure 3) and suitable for flow cytometry analysis. As also pointed out by the reviewer, there is no universal marker for fibroblasts, which complicates direct comparisons to other reports. The data represented in figure 3 can also be plotted for the EpCAM⁻/CD49f⁻/CD45⁻/CD31⁻ cells (see below in figure III). This does not significantly alter the outcome, except that more dTomato⁻ cells will be recorded in MMEC transplanted tumors, because now the EMT tumor cells, which lack fluorescence, will be included in the fibroblast population, but the majority of cells (>82%) is Tomato-positive and therefore host-derived rather than tumor cell-derived. In supplemental figure 3 we have shown that Tomato-

negative tumor cells, independent of their EpCAM or CD49f expression, do not contract collagen *in vitro*, suggesting they do not function as CAFs.

For our single cell transcriptomics we decided to perform a negative selection, allowing us to capture more cells and potentially also more rare CAF subtypes. In this experiment, EMT tumor cells, or 'contaminating cells' from the FACS procedure have been excluded from analysis based on their marker expression (CD31/EpCAM/Ker8/Ker14/CD45) and/or loss or overexpression of the genes known to drive tumorigenesis in our models (*Pten*, *Pik3ca*, *Brca1*, *Trp53*). We have provided additional information in the manuscript for the fibroblast markers used (page 5 in yellow) and added the suggested reference and discussed the paper (page 15, yellow highlighted).

Figure III

Figure III: Nearly all CAFs in TNBC and ILC originate from mammary tissue-resident fibroblasts. A-C) Re-analysis of data in manuscript figure 2. In this analysis fibroblasts were defined as EpCAM-/CD31-/CD45- (WEPtn and WEH1047R, no CD49f was used for analysis of these tumors) or EpCAM-/CD49f-/CD31-/CD45- (WB1P and WB1P-Myc). Panel A shows results from MMEC transplantations, panel from BM transplantations and panel C from whole MG transplantations.

12) The statement “Since these findings contradict ...(ref. 14)” needs to be clarified. In which way are the findings contradictory? Ref. 14 finds that BM-CAFs do not express PDGFR α as opposed to resident CAFs and thus, can distinguish the two origins.

We believe our findings are contradictory to those described by Raz *et al.* because if we use a negative selection strategy to define fibroblasts (EpCAM-/CD49f-/CD45-/CD31-), which should include the PDGFR α -negative CAFs described by Raz *et al.*, we do not observe significant fibroblast recruitment from the bone marrow, as nearly all (>97%) fibroblasts are Tomato-negative (see figure III above). Furthermore, in our experiments with the EPtn;mTmG mice transplanted with FVB/n bone marrow, we also found that the majority of PDGFR β - CAFs were Tomato+ and therefore originated from the host, rather than the transplanted bone marrow.

13) Suppl. Fig. 5 D and E and p. 8: “Again, BM-MSCs were not the main contributors to the population of CAFs, as the majority of the CAFs in the resulting mammary tumors were tdTomato-positive”. Fig. 4D shows the tdTomato expression among each of the CD45+, CD31+ and PDGFR β + populations. Is it the higher frequency of tdTomato+ among PDGFR β + cells that leads to this conclusion? If the transplanted MSCs lacked fluorescence, where do the tdTomato+ /CD45+ cells (approximately 20%) come from?. Please clarify and rewrite. Also, an analysis similar to the one done for the normal gland in Suppl. Fig. 6 B would be helpful.

The reviewer is correct. We indeed verified that BM-MSCs did not contribute to the population of CAFs, because the majority of the PDGFR β + CAFs expressed Tomato and therefore were not derived from the Tomato-negative bone marrow that was transplanted in these EPtn;mTmG mice. We acknowledge that positive selection of PDGFR β may not capture all CAFs present in a tumor and therefore added the data of the PDGFR β -negative CAFs to the graph in supplemental figure 5E. The majority of PDGFR β -negative CAFs (>85%) were Tomato-positive and therefore host-derived, rather than bone marrow-derived. In contrast to the data from our other transplantation studies displayed in figure 2, we could exclude EMT tumor cells from the PDGFR β - population by their GFP expression in this experiment (see supplemental figure 5D,E).

We understand that the display of the proportion of Tomato-positive and Tomato-negative CD31+ and CD45+ cells graph in Supplemental figure 5D (5E in revised version) may cause some confusion. The populations displayed in this graph are the intratumoral endothelial and immune cell populations. This is not data derived from flow cytometry analysis of the transplanted bone marrow. By lethal irradiation and subsequent bone marrow transplantation it is impossible to eliminate all tissue-resident CD45+ immune cells, as some subtypes of mature immune cells are very resistant to irradiation (tissue-resident macrophages and mast cells for example). Nevertheless, analysis of the bone marrow in this experiment showed full reconstitution with the donor FVB/n bone marrow as nearly all MSCs lacked Tomato expression. The data of this analysis was added to the manuscript (see supplemental figure 5G). As requested, we added the analysis on the distribution of the individual cell populations of these tumors and displayed them in supplemental figure 5D.

14) The presentation of results in Fig. 3 is of major concern. Cluster 2 is said to represent iCAFs and cluster 4 myCAFs, but who is who in Fig. 3F and 3G? The iCAF signature now corresponds to cluster 4 and the other way around. Until this is fixed, it is not possible to assess whether the following results presented are correct ! It is claimed that there is a mix of CAFs with iCAF and myCAF signatures in the tumors, but is this really true then?

We apologize for this mistake. The reviewer is absolutely correct; the labels were switched. We have corrected this in the revised manuscript. The gene signature displayed in original figure 3F was the myCAF signature and the data in figure 3G was the iCAF signature. We optimized our iCAF and myCAF gene signatures as per request of reviewer 2 and 3 and moved this data to supplemental figure 8G of the revised manuscript.

15) The authors subsequently focuses on CD26, and say that “The presence of two distinct fibroblast populations that could be distinguished from each other by CD26 expression has previously been reported in normal human and mouse mammary tissue (refs. 16, 31)”, p. 12. Firstly, it is somewhat understated as the two previous studies found an immune-related signature in CD26+ fibroblasts, and that Dpp4+ Fib_0 represented “INFLAMMATORY RESPONSE”, so to claim CD26+ to contribute iCAFs as “our” hypothesis is not entirely appropriate. Secondly, how do the cluster 3 CD26+ relate to these previously described iCAF (?) populations? In the present study – provided that Fig. 3C is correct of course...- cluster 2 rather than cluster 3 represents iCAFs. At the very least this should be reflected upon and discussed.

We have included in our manuscript that others also found an immune-related signature associated with CD26+ NFs (see page 16, yellow highlighted). However, an inflammatory response in normal fibroblasts should not be confused with iCAFs, which are inflammatory fibroblasts in a cancer setting. Our data suggests that cluster 3 are the iCAFs, considering that this cluster emerges during tumor development, in contrast to cluster 2 (CD26+ NFs), which is already present in the normal, non-tumor-bearing, mammary gland and should therefore be considered normal fibroblasts. In our data, we do observe a shift of cluster 2 (CD26+ NFs) towards cluster 3 (iCAFs) as tumor progress, suggesting that this cluster changes when tumors develop and indicate that part of this cluster (indicated by the arrow in figure IV below) could contribute to the iCAF population.

The methodology used for clustering was based on the combined dataset (all time points together), to clearly visualize the changes over time, but this does cause the algorithm to fit the cells of individual time points to clusters defined on the entire dataset. Analysis of each of the time points individually creates unique UMAPs for each sample, making it hard to interpret relationships between clusters across time points. The overlap in gene expression between CD26+ NFs (cluster 2) and iCAFs (cluster 3) and the observation that a pancreatic iCAF signature (supplemental figure 9) also marker CD26+ NFs, suggests that these two fibroblast subtypes (one found in tumor setting and the other in normal mammary gland) are related. In addition, clustree analysis (see revised supplemental figure 5F) also shows close proximity of CD26+ NF and iCAF clusters, again suggesting a functional relationship between these fibroblast subtypes.

Figure IV

Figure IV: CD26+ NF cluster (cluster 3) changes as tumors progress. Fibroblasts were isolated from control, non-tumor bearing mice and ILC-tumor bearing mice (Cdh1F/F;PtenF/F mice) at indicated time points after tumor initiation. The CD26+ NF cluster (cluster 3) is outlined in the different time point samples to indicate that this populations shifts towards the iCAF cluster (cluster 2) as tumors progress (arrow).

16) Fig. 4A and D “..confirming the absence of CD26 in myCAFs and low CD26 expression in iCAFs”. So now, iCAFs express low CD26. What is low compared to – og +? And then there is a shift back to CD26- and CD26+. This is very confusing!

We agree that this sentence may cause confusion. On mRNA level we observe less *Dpp4* expression in iCAFs compared to CD26+ NFs (see Figure V below). At protein level, we observe only a minor decrease in CD26 expression in CAFs compared to NFs as determined by flow cytometry, and therefore these cells are still considered CD26+ (see Figure V below). To avoid any confusion we rephrased the sentence on page 10 to: ‘...confirming the absence of CD26 in myCAFs and CD26 expression in iCAFs within the single-cell transcriptomics dataset’.

Figure V

Figure V: Dpp4 (CD26) expression in normal fibroblasts and CAFs. A) Violin plot of *Dpp4* expression in single-cell RNAseq dataset of fibroblasts over the course of tumor development (see manuscript figure 3A). B) CD26 protein expression among fibroblasts in normal mammary gland and ILC (WEPTn-derived tumor) by flow cytometry. Fibroblasts were defined as EpCAM-/CD45-/CD31-.

17) The stroma of ILC and TNBC comprises both iCAF and myCAF signatures.

P. 15: “Together, these findings indicate that CD26+ are at the origin of pro-tumorigenic CAFs.” Again, others have found that CD26low normal breast fibroblasts have more genes in common with tumor stroma than CD26high fibroblasts (ref. 31), and it has been argued that in pancreatic ductal adenocarcinoma, CD26 expression indicates phenotypic states (Hutton

et al. Cancer Cell, 2021). There is a need for further distinction, reflection and discussion in relation to mouse versus human. In the human gland CD26+ fibroblasts surround the ducts and CD26- surround the terminal duct lobular units (which the mouse does not have), and duct- and TDLU-derived breast carcinomas are known to exhibit unique histological appearances and clinical outcomes (Tabar *et al. Breast Cancer, 2014*).

The reviewer is indeed correct that others have shown more similarity between CD26- NFs and tumor stroma than between CD26+ NFs and tumor stroma. However, it is important to realize that this comparison was based on data derived from bulk RNA sequencing, in which no distinction was made between iCAFs and myCAFs, and we have shown that, across our mouse models, there is an increase in CD26- CAFs and a decrease in CD26+ CAFs over the course of tumor development (see supplemental figure 11). This could mean that the overlap between CD26- NFs and tumor stroma could merely be explained by the relative abundance of one population over the other in the tumor setting.

Hutton *et al.* suggests that CD26 expression in pancreatic cancer-associated fibroblasts indicate phenotypic states rather than defined lineages based on the graded expression of CD26 within these tumors, but no comparisons between CD26+ and CD26- fibroblasts were done in that study. In addition, a comparison between mammary and pancreatic fibroblasts should be based on functionality rather than single markers.

Human and mouse mammary glands are indeed different in histological appearance and in human breast tissue the CD26+ NFs are located around ducts whereas CD26- NFs are located around lobules. From our whole mount analysis we did not observe a clear localization of CD26+ NFs in adult mouse mammary glands, as CD26+ NFs were present both at terminal end buds and around ducts. We have highlighted these discrepancies between human and mouse in the discussion (see page 15-16, in yellow).

Reviewer #2, expertise in CAFs/TME, mouse models and sc-RNAseq:

The study by Houthuijzen et al was designed to uncover the origin of cancer-associated fibroblasts in mammary tumors using advanced GEMM models of breast cancer. Based on various transplantation experiments, the authors conclude that CAFs originate from resident mammary gland fibroblasts. Through single cell transcriptomics, subsets of normal fibroblasts with or without expression of CD26 are identified to give rise to iCAF and myCAF, respectively. Taken together, the study sheds light on the complexity of breast CAFs and may suggest a path towards precision targeting. While the study is well executed technically, a number of issues remain to be resolved before publication is warranted.

1. The authors are commended on the design of the illuminating transplantation experiments. However, the assessment of CAF abundance by FACS should be complemented by visualization in tissue. In particular, the conclusion by the authors that epithelial cells that have gone through an EMT do not constitute a significant part of the CAF pool may be flawed, since the negative selection marker CD49f has been proven to be expressed by EMT-derived CAFs (see e.g. <https://doi.org/10.1002/stem.791> and <https://doi.org/10.1038/s41467-018-07582-3>). To support this conclusion, the authors should provide alternative evidence.

In the revised manuscript we have provided stainings of iCAFs and myCAFs in all four mouse models used throughout the paper. Analysis of general CAF markers in our single-cell dataset showed differential expression of S100a4 (Fsp1) and Acta2 (SMA) in iCAFs and myCAFs. Immunofluorescent staining of iCAFs and myCAFs using antibodies against FSP1 and SMA showed expression of these markers by fibroblasts located in the stromal compartment of tumors. Additionally, little to no overlap was seen between FSP1 and SMA in our mouse models. This data was included in supplemental figure 7B,C and in the text on page 7-8 of the revised manuscript (highlighted in blue).

The reviewer is in part correct. In the WB1P and WB1P-Myc models we have defined fibroblasts as being negative for CD49f. However, for the ILC models we did not use CD49f as an exclusion marker (see gating strategy in supplemental figure 1), since these tumors arise from luminal (EpCAM+) cells, meaning any EpCAM-/CD49f+ cells would have been included in the fibroblast gate within the ILC models (WEPTn and WEH1047R) and in MMEC transplantations performed with these models we do not observe any contribution of Tomato-negative, tumor-derived cells to the population of CAFs. Within the WB1P and WB1P-Myc models we indeed used the combination of EpCAM (luminal marker) and CD49f (basal marker) to define mammary tumor cells, which are positive for both markers.

We thank the reviewer for bringing to our attention the publication that showed that EMT-derived CAFs express CD49f. We have re-analyzed our MMEC transplantation data of the WB1P and WB1P-Myc tumors to determine the percentage of EpCAM-/CD49f+ cells that could potentially be EMT-derived CAFs. We found that the percentage of EpCAM-/CD49f+ cells is low in these tumors (1.8% and 1.7% of non-endothelial and non-immune cells in WB1P and WB1P-Myc tumors, respectively), indicating that if these are indeed CAFs, they would on average account for 12.9% and 12.5% of all stromal cells in WB1P and WB1P-Myc respectively (Tomato+/CD45-/CD31- cells and Tomato-/CD31-/CD45-/EpCAM-/CD49f+ cells combined). To test if these cells could potentially be CAFs we sorted and cultured EpCAM+/CD49f+ cells, EpCAM-/CD49f- cells and EpCAM-/CD49f+ cells from MMEC transplanted WB1P and WB1P-Myc tumors. Assessment of their ability to contract collagen in culture showed us that the EpCAM-/CD49f+ cells were unable to contract collagen,

suggesting that these cells are not bonafide CAFs. This data, including a quantification of the collagen contraction of all tumor cell populations (EpCAM+/CD49f+, EpCAM-/CD49f+ and EpCAM-/CD49f-) has been added to the manuscript in supplemental figure 3F-G and in the text on page 5 (blue highlighted). To determine if the EpCAM-/CD49f+ cells were tumor cells or non-malignant cells that persisted after transplantation, we performed PCR analysis for presence of floxed or deleted *Trp53* and *Brca1* alleles. All EpCAM-/CD49f+ cultured displayed deleted *Trp53*, indicating that these are bonafide tumor cells. The results of these experiments and the reference of Sarrio *et al.* Stem Cells 2012 have been added to the manuscript (see page 5).

2. Cluster 4 CAFs are designated as myCAF by the authors based on GO analysis of a 25-gene signature of differentially expressed genes. However, the GO categories identified are mostly related to ECM, not contraction. The authors should demonstrate the superior contractile ability of Cluster 4 myCAF over Cluster 2 iCAF. Also, differential gene expression analysis between all four clusters (presumably two NF clusters and 2 CAF clusters) may hide functional relationships between the clusters since the algorithm is looking for unique properties of each cluster. The authors should explore other clustering resolutions and visualize this with a clustree in order to further inform the differential gene expression analysis.

As suggested by the reviewer, we isolated primary iCAFs and myCAFs tested their ability to contract collagen *in vitro*. First, we investigated other cell surface markers, in addition to CD26, that would discriminate between iCAFs and myCAFs and allow for FACS sorting of live cells that could be maintained and tested *in vitro*. Using our single-cell dataset of WEPTn-derived CAFs, we observed differential expression of *Cd34*, *Cd55* and *Dpp4* (CD26) between iCAFs and myCAFs (see Figure VI below and in revised figure 3F,G). Flow cytometry analysis on primary ILCs derived from our WEPTn model, confirmed the presence of CD34- and CD34+ CAFs. As expected from the single-cell transcriptomics data, a large portion of the CD34+ CAFs co-expressed CD55 and CD26, whereas CD34- CAFs expressed less CD55 and CD26. Isolation of CD34- and CD34+ CAFs and assessment of selected iCAF and myCAF markers by Q-PCR and western blot showed that CD34- CAFs expressed myCAF markers whereas CD34+ CAFs expressed iCAF markers (see Figure VI below, data also added to revised manuscript figure 3H and supplemental figure 7D). Next, iCAFs and myCAFs were sorted from WEPTn tumors and tested for their ability to contract collagen *in vitro*. We observed superior collagen contraction by myCAFs compared to iCAFs. The results of these experiments are added to the manuscript in figure 3J and in the text on page 8 (blue highlighted).

We agree with the reviewer that additional clustering resolutions may provide insight into the functional relationships between the fibroblast clusters. Clustree analysis on our single-cell dataset shows a clear and early separation of myCAFs from the other fibroblast clusters regardless of the resolution chosen to analyze the data. The iCAF cluster and CD26+ NF cluster appear to be related and display cluster instability and cross-over of clusters at higher resolutions (see Figure VII below and revised figure 5F). The CD26- NF cluster, like the myCAFs, appears to be more separate from other clusters. This clustree analysis further strengthens our *in vitro* and *in vivo* findings that CD26+ NFs and iCAFs are functionally related. In addition, the clustree analysis confirms our choice of a 0.3 resolution for analysis, which we have based on the PCA silhouette score (see Figure VII).

Figure VI

Figure VI: iCAFs and myCAFs differ in CD34 expression and collagen contracting properties. A) Violin plots showing the relative gene expression of *Cd34*, *Cd55* and *Dpp4* (CD26) in cluster 2: iCAFs and cluster 4: myCAFs within our single-cell dataset of fibroblasts isolated from WEPTn mice (original data in Figure 3 of manuscript). B) Flow cytometry strategy to isolated iCAFs and myCAFs based on CD34 expression. CAFs were defined as EpCAM-/CD45-/CD31- cells. Consistent with gene expression analysis in panel A, the CD34+ CAFs co-expressed CD26 and CD55. C) Q-PCR analysis of primary iCAFs and myCAFs isolated from WEPTn-derived tumors for iCAF and myCAF markers. D) Western blot analysis of freshly isolated iCAFs and myCAFs from WEPTn-derived ILCs

for selected iCAF (orange) and myCAF (blue) markers. E) Quantification and representative image of collagen contraction assay with WEPTn-derived tumor cells, iCAFs and myCAFs.

Figure VII

Figure VII: Clustree analysis reveals functional relationship between CD26+ NF and iCAF clusters. A) Clustree analysis at various resolutions. Box indicates resolution (0.3) and labeling used throughout the manuscript to analyze the data resulting in four fibroblast clusters. B) myCAF gene signature visualized on clustree. C) iCAF gene signature visualized on clustree. D) UMAP representation of our data at various resolutions (all time points combined). E) PCA silhouette score used to determine optimal resolution to analyze the single-cell dataset (0.3 gave highest PCA score).

3. In Fig. 3C, the label for the Y-axis is absent. What do the different sets of time points represent? A matrix with the expression of each set of top 10 DEGs in all CAF clusters would have been helpful. Also, the iCAF/myCAF cluster signatures are not very convincing when applied to other datasets. How would they look visualized on the original feature plot? And on other single cell datasets for breast cancer?

We have extended the Y-axis labeling in Figure 3C to include the clusters and time points more clearly. Figure 3C is indeed a matrix plot of the top-10 differentially expressed genes

(based on Rank-score) in each of the fibroblast clusters. The time points on the Y-axis reflect the same time points and samples as shown in Figure 3A. The sets of time points reflect the clusters over time.

The iCAF and myCAF signatures extracted from our single-cell dataset have been improved. Initially we selected the top-25 differentially expressed genes solely based on Rank-score in a binary one-vs-rest cluster comparison. Since this cut-off of top-25 is relatively arbitrary and selection based on Rank-score does not always reflect the most differentially expressed genes, we changed our approach and based the improved iCAF and myCAF gene signatures on a one-vs-rest cluster comparison with a $\log_2FC > 1.5$, $FDR < 0.05$ and a Rank-score > 20 to select for highly expressed genes. Additionally, the genes meeting these criteria were filtered for genes that in a direct comparison of iCAF vs myCAF, (or vice versa) showed a $\log_2FC > 1.5$, $FDR < 0.05$ and Rank-score > 20 . This resulted in an iCAF gene signature of 14 genes and a myCAF gene signature of 53 genes (see supplemental table 1 of revised manuscript). In Figure VIII below, the old and new iCAF and myCAF signatures are shown on the original dataset. The new iCAF and myCAF signatures are also shown on the original dataset in supplemental figure 7A of the revised manuscript. In addition, we have performed single-cell transcriptomics on ILC tumors isolated from WEPTn (n=3) and H1047R (n=3) mice. This dataset reflects the entire tumor microenvironment and assessment of the iCAF and myCAF gene signatures in this dataset shows iCAF and myCAF gene expression specifically in the fibroblast clusters and is absent in other cell types. The results of this analysis is added to the manuscript in figure 4A-E and in the text on page 8 (blue highlighted).

We agree with the reviewer that our iCAF and myCAF signatures do not appear visually convincing in the heatmaps of the laser microdissected human datasets (ILC in Figure 3F and G and IDC in supplemental figure 7B and C of original manuscript). Nevertheless, the new iCAF and myCAF signature scores are significantly enriched in tumor stroma compared to tumor epithelium in these datasets (data added to revised manuscript in figure 4K,L). However, it is important to realize that laser microdissection provides an enrichment of a target cell population, not a purification, meaning that some contaminating cells like endothelial cells or immune cells could be present in samples allocated as tumor stroma or tumor epithelium. Nevertheless, this dataset is unique to ILC research as it provides a separation of tumor stroma and tumor epithelium, in contrast to publically available bulk transcriptomics datasets.

Since we have found in our mouse models that the iCAF and myCAF signatures extracted from ILC-derived CAFs also apply to TNBC-derived CAFs, we interrogated the recently published single-cell dataset of human breast cancer by Pal *et al.* (EMBO J, 2021) to determine how our iCAF and myCAF signatures perform on this human dataset. As expected the iCAF and myCAF signatures locate to the fibroblast cluster within this dataset and re-clustering of the fibroblasts shows iCAF and myCAF clusters (data added to manuscript in figure 4M-Q and supplemental figure 8I,J and in text on page 9). Taken together, our iCAF and myCAF gene signatures show validity on additional mouse and human datasets.

Figure VIII

Figure VIII: UMAP representation of old and new iCAF and myCAF gene signatures. Old signatures were based on top-25 of differentially expressed genes in a binary one-vs-rest comparison based on Rank-score. New signatures are based on one-vs-rest binary comparison with the following criteria: $\text{Log}_2\text{FC} > 1.5$, $\text{FDR} < 0.05$ and $\text{Rank-score} > 20$ with additional filtering comparing iCAFs vs myCAF with the following criteria: $\text{Log}_2\text{FC} > 1.5$, $\text{FDR} < 0.05$ and $\text{Rank-score} > 20$.

4. In the cell culture experiments using isolated CD26⁻ and CD26⁺ fibroblasts, the authors need to demonstrate that these truly represent stable lineages by assessing CD26 expression over time. Also, while gene expression signatures appear to be regulated in co-culture experiments, these say nothing about functionality. Are CD26⁻ fibroblasts conditioned by ILC cells superior in contractile abilities? Similarly, do CD26⁺ fibroblasts modulate immune cell function following tumor cell conditioning?

We agree with the reviewer that assessment of CD26 expression of the cultured fibroblasts may provide information on stability in culture. By flow cytometry analysis we found that CD26 expression is quickly lost in culture (see Figure IX and in supplemental figure 14I,J). Despite the rapid loss of membrane expression of CD26 on CD26⁺ NFs, we did observe differential response of CD26⁻ and CD26⁺ NFs in various assays. As shown in supplemental figure 13, CD26⁻ and CD26⁺ NFs respond differently to the same tumor CM. Furthermore, in our organotypic invasion assays and recruitment assays in figure 6D and 8D-F, CD26⁻ and CD26⁺ NFs differ in their ability to induce tumor cell invasion and recruit monocytes, suggesting that although CD26 expression may not be maintained in culture, functional differences between cultured CD26⁻ and CD26⁺ NFs do remain. *In vivo*, fibroblasts remain in a quiescent state and by taking them into culture and inducing them to proliferate, these fibroblasts become activated. Others have also shown that culture conditions of fibroblasts can dictate their behavioral outcome (Biffi *et al.* Cancer Discovery 2019). Therefore we emphasized in our manuscript that *in vivo* validation of experiments with cultured NFs is warranted (see page 12 (in blue) and page 17 (in yellow)).

Collagen contraction assays in the absence or presence of tumor cells showed us that both cultured CD26⁻ and CD26⁺ NFs are able to contract collagen and that this is enhanced in the presence of tumor cells. Collagen contraction assays with freshly isolated CD26⁻ and CD26⁺ NFs shows that these cells lack collagen contractibility, consisted with a quiescent phenotype *in vivo*. Co-cultures of tumor cells with freshly isolated CD26⁻ and CD26⁺ NFs induced collagen contractibility, indicating that this behavior is induced by tumor

cells. CD26- NFs appear to be superior in collagen contraction compared to CD26+ NFs but only in the presence of tumor cells. Important to note it that this difference was not statistically significant (see Figure X below and in figure 7L of the revised manuscript).

We have shown in our manuscript that conditioning of CD26+ NFs with tumor cells can enhance the recruitment of CD11b+ monocytes in transwell assays and that this effect is mediated by CXCL12 secretion (see figure 8C,D).

Figure IX

Figure IX: culturing of CD26- and CD26+ NFs rapidly reduces CD26 expression. A) Representative plots of CD26 protein expression on CD26- NFs (top row) and CD26+ NFs (bottom row) determined by flow cytometry directly after isolation from mammary glands (purity check after sorting) or after 3, 5 or 7 days in culture. B) Quantification of representative data in panel B (n=4 independent experiments with similar outcome).

Figure X

Figure X: collagen contraction of primary and cultured CD26- and CD26+ NFs. A) Quantification of collagen contraction of primary (freshly isolated, FACSsorted) CD26- and CD26+ mammary fibroblasts plated alone or in combination with WEPTn-derived tumor cells or cultured (1 week) CD26- and CD26+ NFs alone or in combination with WEPTn-derived tumor cells (n=4 experiments with similar outcome). Level of contraction was determined compared to no-cells control. B) representative image of collagen contraction assay with primary and cultured CD26- and CD26+ NFs.

5. The fact that the authors need to use alternative culturing conditions to induce CD26- and CD26+ fibroblasts into myCAFs and iCAFs, respectively, detracts from the credibility of the proposed model. *In vivo* co-transplantation experiments of differentially labeled CD26- and CD26+ fibroblasts and subsequent analysis of the fate of these cells as they convert into CAFs would be more illuminating and necessary to support the model.

We agree with the reviewer that the proposed experiment would provide a definitive answer regarding the predispositioning of normal fibroblasts to become distinct CAF subtypes. However, these experiments are technically very challenging because non-transformed cells like fibroblasts do not survive long-term after co-transplantation *in vivo*. Additionally, we have shown that CD26- and CD26+ NFs do not retrain their distinct *in vivo* gene expression profiles in culture, meaning that the proposed experiment may fail due to the significant changes in phenotypes. Nevertheless, we have attempted to perform the suggested experiment using mammary fat pad injections of WEPtn-derived tumor cells either alone or in combination with Tomato+ CD26- NFs and/or GFP+ CD26+ NFs. By combining 10,000 tumor cells with 500,000 NFs (either CD26- alone, CD26+ alone or mixed in 1:1 ratio) we maximized our chance of re-isolating enough NFs for analysis from palpable tumors two weeks later. Consistent with others we observed enhanced tumor growth in the animals co-injected with tumor cells and fibroblasts compared to mice injected with tumor cells alone (see Figure XI below). Flow cytometry analysis of the resulting tumors revealed that less than 0.05% of the injected cells consisted of fibroblasts, resulting in such low cell counts, that any phenotypical analysis of these cells was technically not possible. In the mice co-injected with CD26- and CD26+ NFs together with tumor cells, we observed a shift in balance between CD26- and CD26+ fibroblasts in established tumors. More CD26-/Tomato+ fibroblasts were present compared to CD26+/GFP+ fibroblasts, consistent with a larger fraction of myCAF compared to iCAFs in our GEMMs. However, this difference was not observed when tumor cells were injected with either CD26- or CD26+ NFs alone.

Figure XI

Figure XI: Co-injection of tumor cells with CD26- and CD26+ NFs enhance tumor growth, but fibroblasts do not persist *in vivo*. A) WEPtn-derived tumor cell line 13-MCB-17 was injected in the mammary fatpads of adult mice (n=2 mice, two glands injected per mouse, 10.000 cells per gland) either alone or in combination with CD26- NFs (500.000 cells), CD26+ NFs (500.000 cells) or a combination of CD26- (250.000 cells) and CD26+ NFs (250.000 cells). CD26- NFs constitutively expressed Tomato and CD26+ NFs constitutively expressed GFP. Tumors were harvested after 14 days and weighted. B) After harvesting of the tumors, they were processed to single cell suspension and sorted for tumor cells, Tomato+ (CD26-) NFs and GFP+ (CD26+) NFs. Relative abundance of each cell type is shown, CD31+ endothelial cells and CD45+ immune cells were excluded. C) Representative flow cytometry plots of tumor harvested from mouse injected with tumor cells alone (top panel) and tumor harvested from mouse co-injected with tumor cells, CD26- NFs and CD26+ NFs (bottom panel).

Reviewer #3, expertise in CAFs and sc-RNAseq:

In their manuscript, Houthuijzen and colleagues identify normal tissue fibroblasts as the main source of CAFs in both invasive lobular breast cancer and triple negative breast cancer models. They further characterize fibroblast heterogeneity in normal tissue and in the breast cancer TME and identify CD26 as a marker of two subsets of fibroblasts that are predicted to give rise to distinct populations of CAFs. Based on *in vitro* modelling and *in vivo* experiments the authors suggest that CD26+ fibroblasts primarily give rise to inflammatory CAFs, while CD26- fibroblasts develop into myofibroblastic CAFs.

Overall, the manuscript is well written and logically structured. The *in vivo* experiments which establish normal mammary tissue-resident fibroblasts as the main source of CAFs in breast cancer are novel and of interest to the field. The key challenges I see with the current version of the manuscript are:

Major

1) The En1-Cre;mTmG lineage tracing system which is intended to *in vivo* trace CD26+ and CD26- fibroblasts is not suited for this purpose due to the non-specific expression of En1. Therefore, the authors rely on CD26 protein measurements to infer CAF origin *in vivo* and not the reporter. This is a) very time intensive to follow for the reader without any benefit of the reporter system and b) insufficient evidence for the conclusion that “*in vivo* tracing and *in vitro* studies revealed the transition of CD26+ and CD26- NF populations into inflammatory CAFs (iCAF) and myofibroblastic CAFs (myCAF), respectively”. Additionally, in the *in vitro* systems the observed CAF subtype was highly dependent on the culture system used and therefore these data do also not allow a direct conclusion on the proposed lineage relationships.

We appreciate the reviewer’s concern. Indeed, the En1-Cre;mTmG model did not allow unambiguous tracing of CD26+ NFs in the mammary gland since a significant portion of the CD26+ NFs is not labelled by GFP and therefore not a descendant of the Engrailed1 lineage. By isolating and analyzing all four populations we were able to establish that En1-derived CD26+ or CD26- NFs did not differ from EN1- CD26+ or CD26- NFs in either control or tumor-bearing mice. We have rephrased our concluding sentence in the abstract to: ‘*in vivo* and *in vitro* studies suggests the transition of CD26+ and CD26- NF populations into inflammatory CAFs (iCAF) and myofibroblastic CAFs (myCAF), respectively’. In addition, we removed the term ‘lineage tracing’ from the manuscript.

2) The authors attempt to untangle CAF origin and programs specifically induced in these cells in a tumor setting. Yet, their signatures for iCAF and myCAF remain convoluted and a mixture of origin and induced programs. This convoluted signature [derived from end-stage WB1P and WB1P-Myc tumors] biases the analyses for example in figure 6B: Half of the genes shown visually look already enriched in CD26+ vs CD26- normal fibroblast. As such, the conclusion “As expected, the iCAF gene signature was enriched in the CD26+ NFs cultured in tumor CM” is (visually) true, but partially because of the convoluted baseline expression program and not because TCM induced the expression of these genes in one or the other population (In fact, some of the iCAF genes seem lost with TCM in CD26+ fibroblasts). I would suggest the authors improve their CAF signatures using the single-cell

data from their time course in ILC-tumor bearing mice to be able to make more reliable statements on genes induced by tumor conditioned media in CD26+/- cells respectively.

We agree with the reviewer that our iCAF and myCAF signatures could be improved. In the revised manuscript, we changed our approach to extract new and improved iCAF and myCAF gene signatures from our data (see supplemental table 1). Just to clarify, both iCAF and myCAF gene signatures (old and new) were extracted from the time course in ILC-tumor bearing mice and not derived from end-stage WB1P and WB1P-Myc tumors. Our previous approach to generate the signatures was based on Rank-score in a binary one-vs-rest cluster comparison (top-25). This does not always reflect the most differentially expressed genes and therefore we changed our approach and based the new iCAF and myCAF gene signatures on a one-vs-rest cluster comparison with a $\log_2FC > 1.5$, $FDR < 0.05$ and a Rank-score > 20 . Additionally, the genes meeting these criteria were filtered for genes that in a direct comparison of iCAF vs myCAF, (or vice versa) showed a $\log_2FC > 1.5$, $FDR < 0.05$ and Rank-score > 20 . This resulted in an iCAF gene signature of 14 genes and a myCAF gene signature of 53 genes (see supplemental table 1). In Figure VIII of this rebuttal, the old and new iCAF and myCAF signatures are shown on the original dataset. The data in figure 6 of the original manuscript (figure 7I,J and supplemental figure 14A-H of the revised manuscript) and newly added data in the revised manuscript (figure 4, supplemental figure 8) has been updated with these new iCAF and myCAF gene signatures.

Although these iCAF and myCAF signatures perform well on *in vivo* data (mouse models and human single-cell datasets, see figure 4), in our *in vitro* experiments we still observe expression of both iCAF and myCAF genes in cultured CD26- and CD26+ NFs, even without tumor cell conditioning (see supplemental figure 14). Although functional differences exist between cultured CD26- and CD26+ NFs (see figure 6 and figure 8D-E of revised manuscript) we found that CD26 marker expression, as measured by flow cytometry, was not maintained in cultured CD26+ NFs (see Figure IX), suggesting that cultured CD26- and CD26+ NFs do not fully recapitulate their *in vivo* counterparts. For this reason, we moved the results from our tumor-cell conditioning of NFs by tumor cell CM in 2D or tumor cell co-cultures in collagen to the supplementary data (supplemental figure 14A-G). Furthermore, we emphasized in the revised manuscript that caution is warranted when drawing conclusions from assays with *in vitro* cultured NFs and state that *in vivo* validation is needed. We have added the data showing loss of CD26 expression *in vitro* to the manuscript (supplemental figure 14) and highlighted that *in vitro* results are not directly conclusive for the *in vivo* situation (see page 17, highlighted in yellow).

3) The computational analysis requires more robust statistics and partially lacks replication:

- Some experiments seem to have no replicates ($n=1$), which does not allow to make reliable statements on significance of gene expression changes (e.g Fig 6 E, F not replicated for NF)

The reviewer is indeed correct that for Figure 6E and F (supplemental figure 14F-H in revised manuscript) there is no replicate for the NF only condition, since re-isolation of these cells after culture in collagen was challenging. Therefore, in this analysis we only compared the conditions in which NFs were co-cultured with tumor cells, for which two samples are present, to show increased expression of myCAF markers in CD26- NFs compared to CD26+ NFs.

- In multiple paragraphs the authors make statements about enrichment even though no tests were performed. Often upregulation is inferred “visually”. For example, why are no differential expression tests performed that support the statements on the results in figures 6B,C, E,F, K, L? Similarly: Why is there no statistical test performed for the values in 3H comparing scores in epithelium and stroma?

We have performed statistical tests and the results have been added to the figure panels in supplemental figure 14D,H and figure 7K. In addition, we have improved our statistical analysis throughout the manuscript and indicated the tests used for each panel in the figure legends.

- The rows of the heatmaps are clustered, but no cluster dendrograms are provided in these heatmaps. For some figures the authors might even want to avoid clustering in order to allow comparisons between sub panels more easily (For example it’s hard to compare between 6B and E due to different ordering of the genes).

This is a valid point. We have removed the clustering in these heatmaps and ordered the genes alphabetically for easy comparison between the figure panels.

- For some differential expression comparisons that were performed it is unclear what conditions were compared (“Hierarchical clustering of the top-100 differentially expressed genes” Top differentially expressed between which conditions?)

The data displayed in figure 7H and supplemental figure 13 is not a top-100 differentially expressed genes, but rather a hierarchical clustering of top-100 of most variable genes, but was mislabeled in the manuscript. We have corrected this and also indicated that the scores in these figures are Z-scores (relative expression). We have updated the figure labeling and reference in the manuscript text (see page 11 and 13, yellow highlighted).

- The legends in multiple panels do not have a label for the color range. What is plotted (E.g. 4A, E, 6B,C, E, F, K, L,)?Z-scores? Of what?Log CPM? CPM?

We have updated the labeling in the figure panels. Figure 4A (Figure 5A in revised manuscript) shows average expression (normalized UMIs). Figure 4E (Figure 5E in revised manuscript) shows pseudo time and panels B, C, E, F, K and L of Figure 6 (supplemental figure 14B,C,F,G and figure 7I,J in revised manuscript, respectively) show relative expression (Z-scores).

- Essential information on the single-cell data is missing: Supplemental tables for markers of identified clusters, results of differential expression tests, gene set enrichment analysis, ..

We have included a supplementary table (supplemental table 1) with additional information of the single-cell data. These data include Log2FC, Rank-scores, FDR of the one-vs-rest cluster comparison as well as comparisons between individual clusters and gene ontology analyses.

- The computational single cell methods provided do not allow to fully understand the analyses performed. The only two sentences on downstream analysis after QC/filtering are given as: “Clustering was made using the Leiden algorithm. Trajectory analysis was performed using the PAGA[60].” For none of the analyses performed any parameters are provided, what dimensionality reduction techniques were used, if the analysis was focused on a set of most variable genes, ... All these things would be quite essential to understand how this data has been processed.

We have updated the material and methods section to include more information on how the data was processed and analyzed. This information can be found in the material and methods section on page 28-30.

Minor

Fig 3 F,G: iCAF/myCAF labels flipped; I do not understand what logFC is shown here. Are these z-cores or LogFCs? If logFC compared to what?

We thank the reviewer for noticing that the labels in Figure 3F and G are flipped, we have corrected this mistake. Indeed, the data in Figure 3F and G (supplemental figure 8G in revised version) represent Z-scores and not Log2FC; this has also been corrected.

3I-J: What the authors annotate as CAFs are Rgs5+Acta2+ cells and likely pericytes

The reviewer is absolutely correct; Rgs5 is a marker for pericytes. In all reclustering analyses of *Col1a1*+ fibroblast clusters we have now included a negative selection *Rgs5*-expressing cells to exclude pericytes. The figures have been updated in the manuscript (supplemental figure 8A-F and newly added data in figure 4Q) and this selection for *Col1a1*+/*Rgs5*- cells was mentioned in the manuscript on page 8 (blue highlighted).

SF12 These heatmaps are really hard to interpret. In it looks like CD26 is higher in CD26- fibroblasts with TCM than in CD26+ NT fibroblasts (which seem to have low CD26 expression and CD26 expression increases with conditioned media). Is this data transformed in some kind of way or am I interpreting the figure incorrectly? Seems counterintuitive, especially also when looking at Dpp4 expression comparing CD26+ and CD26- NT fibroblasts. Seems like they express Dpp4 at the same (low) level?

The reviewer is interpreting the data correctly. CD26 expression is downregulated in cultured CD26+ fibroblasts. Additional analyses showed that cultured CD26+ NFs lose their CD26 expression (see figure IX of this rebuttal and supplemental figure 14I,J of the revised manuscript). Although some features may remain, like collagen contractibility and immune cell recruitment, we cannot consider them to be reflecting their *in vivo* counterparts to their full extent. This has been explained in the manuscript on page 12 and 17.

The authors do not attempt to infer relationships between CD26⁺ and – cells in normal tissue. Are CD26⁻ fibroblasts derived from a CD26⁺ cell or vice versa? Can the authors use velocity analyses to make computational predictions?

We believe it is certainly interesting to determine the relationship between CD26⁻ and CD26⁺ NFs and their origins during mammary gland and/or embryonic development. Data from En1-Cre;mTmG mice would suggest, since a portion of both CD26⁻ and CD26⁺ NFs are derived from Engrailed1 positive precursors, that at least part of these cells share a common neuroectoderm ancestor cell during embryonic development. Velocity analysis on our control time point (non-tumor bearing mice, Figure 3A, first panel) suggests that CD26⁻ NFs residing at the border between the clusters of CD26⁻ and CD26⁺ NFs give rise to both populations (see Figure II). However, we believe that our current dataset is not the most suitable for elucidation of the origin of CD26⁻ and CD26⁺ mammary NFs. Single-cell transcriptomic analysis during mammary gland development rather than in adult, steady-state settings or tumor settings should be performed to reliably answer this question and we feel this is beyond the scope of the manuscript.

Reviewers' Comments:

Reviewer #1:

Remarks to the Author:

The authors have responded adequately to the points of criticism raised and revised the manuscript accordingly. This reviewer finds the manuscript ready for publication.

Reviewer #2:

Remarks to the Author:

The authors are commended on a very thorough revision in response to the issues previously raised.

Reviewer #3:

Remarks to the Author:

The authors have successfully addressed my previous concerns and provide a substantially improved manuscript. Congratulations.